# Sharpened Generalization Bounds based on Conditional Mutual Information and an Application to Noisy, Iterative Algorithms

**Mahdi Haghifam**
University of Toronto,
Vector Institute

**Jeffrey Negrea**
University of Toronto,
Vector Institute

**Ashish Khisti**
University of Toronto

**Daniel M. Roy**
University of Toronto,
Vector Institute

**Gintare Karolina Dziugaite**
Element AI

## Abstract

The information-theoretic framework of Russo and Zou (2016) and Xu and Raginsky (2017) provides bounds on the generalization error of a learning algorithm in terms of the mutual information between the algorithm's output and the training sample. In this work, we study the proposal by Steinke and Zakynthinou (2020) to reason about the generalization error of a learning algorithm by introducing a super sample that contains the training sample as a random subset and computing mutual information conditional on the super sample. We first show that these new bounds based on the conditional mutual information are tighter than those based on the unconditional mutual information. We then introduce yet tighter bounds, building on the "individual sample" idea of Bu et al. (2019) and the "data dependent" ideas of Negrea et al. (2019), using disintegrated mutual information. Finally, we apply these bounds to the study of the Langevin dynamics algorithm, showing that conditioning on the super sample allows us to exploit information in the optimization trajectory to obtain tighter bounds based on hypothesis tests.

## 1   Introduction

Let $\mathcal{D}$ be an unknown distribution on a space $\mathcal{Z}$, and let $\mathcal{W}$ be a set of parameters that index a set of predictors, $\ell : \mathcal{Z} \times \mathcal{W} \to [0, 1]$ be a bounded loss function. Consider a (randomized) learning algorithm $\mathcal{A}$ that selects an element $W$ in $\mathcal{W}$, based on an IID sample $S = (Z_1, \dots, Z_n) \sim \mathcal{D}^{\otimes n}$. For $w \in \mathcal{W}$, let $R_{\mathcal{D}}(w) = \mathbb{E}\ell(Z, w)$ denote the risk of predictor $w$, and $\hat{R}_S(w) = \frac{1}{n}\sum_{i=1}^{m} \ell(Z_i, w)$ denote the empirical risk. Our interest in this paper is the *(expected) generalization error* of $\mathcal{A}$ with respect to $\mathcal{D}$,

$$\mathrm{EGE}_{\mathcal{D}}(\mathcal{A}) = \mathbb{E}[R_{\mathcal{D}}(W) - \hat{R}_S(W)].$$

In this work, we study bounds on generalization error in terms of information-theoretic measures of dependence between the data and the output of the learning algorithm. This approach was initiated by Russo and Zou [18, 19] and has since been extended [2, 3, 6, 9, 17, 24]. The basic result in this line of work is that the generalization error can be bounded in terms of the mutual information $I(W; S)$ between the data and the learned parameter, a quantity that has been called the *information usage* or *input–output mutual information of $\mathcal{A}$ with respect to $\mathcal{D}$*, which we denote by $\mathrm{IOMI}_{\mathcal{D}}(\mathcal{A})$. The following result is due to Russo and Zou [18] and Xu and Raginsky [24].

**Theorem 1.1.** $\mathrm{EGE}_{\mathcal{D}}(\mathcal{A}) \le \sqrt{\dfrac{\mathrm{IOMI}_{\mathcal{D}}(\mathcal{A})}{2n}}$.

Theorem 1.1 formalizes the intuition that a learning algorithm without heavy dependence on the training set will generalize well. This result has been extended in many directions: Raginsky et al. [17] connect variants of $\text{IOMI}_\mathcal{D}(\mathcal{A})$ to different notions of stability. Asadi et al. [3] establish refined bounds using chaining techniques for subgaussian processes. Bu et al. [6] obtain a tighter bound by replacing $\text{IOMI}_\mathcal{D}(\mathcal{A})$ with the mutual information between $W$ and a single training data point. Negrea et al. [15] propose variants that allow for data-dependent estimates. See also [1, 2, 4, 9, 10].

Our focus in this paper is on a new class of information-theoretic bounds on generalization error, proposed by Steinke and Zakynthinou [21]. Fix $k \geq 2$, let $[k] = \{1, \ldots, k\}$, let $U^{(k)} = (U_1, \ldots, U_n) \sim \text{Unif}([k]^n)$, and let $\tilde{Z}^{(k)} \sim \mathcal{D}^{\otimes(k \times n)}$ be a $k \times n$ array of IID random elements in $\mathcal{Z}$, independent from $U^{(k)}$. Let $S = (Z_{U_1,1}, \ldots, Z_{U_n,n})$ and let $W$ be a random element in $\mathcal{W}$ such that conditional on $S$, $U^{(k)}$, and $\tilde{Z}^{(k)}$, $W$ has distribution $\mathcal{A}(S)$. It follows that, conditional on $S$, $W$ is independent from $U^{(k)}$ and $\tilde{Z}^{(k)}$. By construction, the data set $S$ is hidden inside the super sample; the indices $U^{(k)}$ specify where. Steinke and Zakynthinou [21] use these additional structures to define:

**Definition 1.2.** The *conditional mutual information of $\mathcal{A}$ w.r.t. $\mathcal{D}$* is $\text{CMI}_\mathcal{D}^k(\mathcal{A}) = I(W; U^{(k)} | \tilde{Z}^{(k)})$.

Intuitively, $\text{CMI}_\mathcal{D}^k(\mathcal{A})$ captures how well we can recognize which samples from the given super-sample $\tilde{Z}^{(k)}$ were in the training set, given the learned parameters. This intuition and the connection of $\text{CMI}_\mathcal{D}^k(\mathcal{A})$ with the membership attack [20] can be formalized using Fano's inequality, showing that $\text{CMI}_\mathcal{D}^k(\mathcal{A})$ can be used to lower bound the error of any estimator of $U^{(k)}$ given $W$ and $\tilde{Z}^{(k)}$. (See Appendix A.) Steinke and Zakynthinou [21] connect $\text{CMI}_\mathcal{D}^k(\mathcal{A})$ with well-known notions in learning theory such as distributional stability, differential privacy, and VC dimension, and establish the following bound [21, Thm. 5.1] in the case $k = 2$, the extension to $k \geq 2$ being straightforward:

**Theorem 1.3.** $\text{EGE}_\mathcal{D}(\mathcal{A}) \leq \sqrt{\frac{2\text{CMI}_\mathcal{D}^k(\mathcal{A})}{n}}$.

This paper improves our understanding of the framework introduced by Steinke and Zakynthinou [21], identifies tighter bounds, and applies these techniques to the analysis of a real algorithm. In Section 2, we present several formal connections between the two aforementioned information-theoretic approaches for studying generalization. Our first result bridges $\text{IOMI}_\mathcal{D}(\mathcal{A})$ and $\text{CMI}_\mathcal{D}^k(\mathcal{A})$, showing that for any learning algorithm, any data distribution, and any $k$, $\text{CMI}_\mathcal{D}^k(\mathcal{A})$ is less that $\text{IOMI}_\mathcal{D}(\mathcal{A})$. We also show that $\text{CMI}_\mathcal{D}^k(\mathcal{A})$ converges to $\text{IOMI}_\mathcal{D}(\mathcal{A})$ as $k \to \infty$ when $|\mathcal{W}|$ is finite. In Section 3, we establish two novel bounds on generalization error using the random index and super sample structure of Steinke and Zakynthinou, and show that both our bounds are tighter than those based on $\text{CMI}_\mathcal{D}^k(\mathcal{A})$. Finally, in Section 4, we show how to construct generalization error bounds for noisy, iterative algorithms using the generalization bound proposed in Section 3. Using the Langevin dynamics algorithm as our example, we introduce a new type of prior for iterative algorithms that "learns" from the past trajectory, using a form of *hypothesis testing*, in order to not "pay" again for information obtained at previous iterations. Experiments show that our new bound is tighter than [14, 15], especially in the late stages of training, where the hypothesis test component of the bound *discounts* the contributions of new gradients. Our new bounds are non-vacuous for a great deal more epochs than related work, and do not diverge or exceed 1 even when severe overfitting occurs.

## 1.1 Contributions

1. We characterize the connections between the $\text{IOMI}_\mathcal{D}(\mathcal{A})$ and $\text{CMI}_\mathcal{D}^k(\mathcal{A})$. We show that $\text{CMI}_\mathcal{D}^k(\mathcal{A})$ is always less than the $\text{IOMI}_\mathcal{D}(\mathcal{A})$ for any data distribution, learning algorithms and $k$. Further, we prove that $\text{CMI}_\mathcal{D}^k(\mathcal{A})$ converges to $\text{IOMI}_\mathcal{D}(\mathcal{A})$ when $k$ goes to infinity for finite parameter spaces.

2. We provide novel generalization bounds that relate generalization to the mutual information between learned parameters and a random subset of the random indices $U_1, \ldots U_n$.

3. We apply our generalization bounds to the Langevin dynamics algorithm by constructing a specific *generalized prior and posterior*. We employ a generalized prior that learns about the values of the indices $U$ from the optimization trajectory. To our knowledge, this is the first generalized prior that learns about the dataset from the iterates of the learning algorithm.

4. We show empirically that our bound on the expected generalization error of Langevin dynamics algorithm is tighter than other existing bounds in the literature.

## 1.2 Definitions from Probability and Information Theory

Let $\mathcal{S}, \mathcal{T}$ be measurable spaces, let $\mathcal{M}_1(\mathcal{S})$ be the space of probability measures on $\mathcal{S}$, and define a probability kernel from $\mathcal{S}$ to $\mathcal{T}$ to be a measurable map from $\mathcal{S}$ to $\mathcal{M}_1(\mathcal{T})$. For random elements $X$ in $\mathcal{S}$ and $Y$ in $\mathcal{T}$, write $\mathbb{P}[X] \in \mathcal{M}_1(\mathcal{S})$ for the distribution of $X$ and write $\mathbb{P}^Y[X]$ for (a regular version of) the conditional distribution of $X$ given $Y$, viewed as a $\sigma(Y)$-measurable random element in $\mathcal{M}_1(\mathcal{S})$. Recall that $\mathbb{P}^Y[X]$ is a regular version if, for some probability kernel $\kappa$ from $\mathcal{T}$ to $\mathcal{S}$, we have $\mathbb{P}^Y[X] = \kappa(Y)$ a.s. . If $Y$ is $\sigma(X)$-measurable then $Y$ is a function of $X$. If random measure, $P$, is $\sigma(X)$-measurable then the measure $P$ is determined by $X$, but a random element $Y$ with $\mathbb{P}^X[Y] = P$ is not $X$ measurable unless it is degenerate. If $X$ is a random variable, write $\mathbb{E}X$ for the expectation of $X$ and write $\mathbb{E}^Y X$ or $\mathbb{E}[X|Y]$ for (an arbitrary version of) the conditional expectation of $X$ given $Y$, which is $Y$-measurable. For a random element $X$ on $\mathcal{S}$ and a probability kernel $P$ from $\mathcal{S}$ to $\mathcal{T}$, the composition $P(X) := P \circ X$ is a $\sigma(X)$-measurable random measure of a random element taking values in $\mathcal{T}$. We occasionally use this notation to refer to a kernel $P$ implicitly by the way it acts on $X$.

Let $P$, $Q$ be probability measures on a measurable space $\mathcal{S}$. For a $P$-integrable or nonnegative measurable function $f$, let $P[f] = \int f \, dP$. When $Q$ is absolutely continuous with respect to $P$, denoted $Q \ll P$, we write $\frac{dQ}{dP}$ for the Radon–Nikodym derivative of $Q$ with respect to $P$. We rely on several notions from information theory: The *KL divergence of $Q$ with respect to $P$*, denoted $\mathrm{KL}(Q \| P)$, is $Q[\log \frac{dQ}{dP}]$ when $Q \ll P$ and $\infty$ otherwise. Let $X, Y$, and $Z$ be random elements, and let $\otimes$ form product measures. The *mutual information between $X$ and $Y$* is $I(X;Y) = \mathrm{KL}(\mathbb{P}[(X,Y)] \| \mathbb{P}[X] \otimes \mathbb{P}[Y])$. The *disintegrated mutual information between $X$ and $Y$ given $Z$*, is[1]

$$I^Z(X;Y) = \mathrm{KL}(\mathbb{P}^Z[(X,Y)] \| \mathbb{P}^Z[X] \otimes \mathbb{P}^Z[Y]).$$

The *conditional mutual information* of $X$ and $Y$ given $Z$ is $I(X;Y|Z) = \mathbb{E}I^Z(X,Y)$.

## 2 Connections between $\mathrm{IOMI}_\mathcal{D}(\mathcal{A})$ and $\mathrm{CMI}_\mathcal{D}^k(\mathcal{A})$

In this section, we compare approaches for the information-theoretic analysis of generalization error, and we aim to unify the two main information-theoretic approaches for studying generalization. In Theorems 2.1 and 2.2 we will show that for any learning algorithm and any data distribution, $\mathrm{CMI}_\mathcal{D}^k(\mathcal{A})$ provides a tighter measure of dependence than $\mathrm{IOMI}_\mathcal{D}(\mathcal{A})$, and that one can recover $\mathrm{IOMI}_\mathcal{D}(\mathcal{A})$–based bounds from $\mathrm{CMI}_\mathcal{D}^k(\mathcal{A})$ for finite parameter spaces.

A fundamental difference between $\mathrm{IOMI}_\mathcal{D}(\mathcal{A})$ and $\mathrm{CMI}_\mathcal{D}^k(\mathcal{A})$ is that $\mathrm{CMI}_\mathcal{D}^k(\mathcal{A})$ is bounded by $n \log k$ [21], while $\mathrm{IOMI}_\mathcal{D}(\mathcal{A})$ can be infinite even for learning algorithms that provably generalize [6]. One of the motivations of Steinke and Zakynthinou was that proper empirical risk minimization algorithms over threshold functions on $\mathbb{R}$ have large $\mathrm{IOMI}_\mathcal{D}(\mathcal{A})$ [4]. In contrast, some such algorithms have small $\mathrm{CMI}_\mathcal{D}^k(\mathcal{A})$. Our first result shows that $\mathrm{CMI}_\mathcal{D}^k(\mathcal{A})$ is never larger than $\mathrm{IOMI}_\mathcal{D}(\mathcal{A})$.

**Theorem 2.1.** *For every $k \geq 2$, $I(W;S) = I(W;\tilde{Z}^{(k)}) + I(W;U^{(k)}|\tilde{Z}^{(k)})$ and*

$$\mathrm{CMI}_\mathcal{D}^k(\mathcal{A}) \leq \mathrm{IOMI}_\mathcal{D}(\mathcal{A}).$$

Next, we address the role of the size of the super-sample in CMI. In [21], CMI is defined using a super-sample of size $2n$ ($k = 2$) only. Our next result demonstrates that $\mathrm{CMI}_\mathcal{D}^k(\mathcal{A})$ agree $\mathrm{IOMI}_\mathcal{D}(\mathcal{A})$ in the limit as $k \to \infty$ when the parameter space is finite.

**Theorem 2.2.** *If the output of $\mathcal{A}$ takes value in a finite set then*

$$\lim_{k \to \infty} \mathrm{CMI}_\mathcal{D}^k(\mathcal{A}) = \mathrm{IOMI}_\mathcal{D}(\mathcal{A}).$$

Combining Theorems 1.3 and 2.2, we obtain

$$\mathrm{EGE}_\mathcal{D}(\mathcal{A}) \leq \lim_{k \to \infty} \sqrt{\frac{2\mathrm{CMI}_\mathcal{D}^k(\mathcal{A})}{n}} = \sqrt{\frac{2\mathrm{IOMI}_\mathcal{D}(\mathcal{A})}{n}}, \tag{1}$$

when the parameter space is finite. Comparing Eq. (1) with Theorem 1.1 we observe that Eq. (1) is twice as large. In Theorem B.1, we present a refined bound based on $\mathrm{CMI}_\mathcal{D}^k(\mathcal{A})$ which asymptotically match Theorem 1.1. The proofs of the results of this section appear in Appendix C.

# 3 Sharpened Bounds based on Individual Samples

We now present two novel generalization bounds and show they provide a tighter characterization of the generalization error than Theorem 1.3. The results are inspired by the improvements on $\text{IOMI}_{\mathcal{D}}(\mathcal{A})$ due to Bu et al. [6]. In particular, Theorem 3.1 bounds the expected generalization error in terms of the mutual information between the output parameter and a random subsequence of the indices $U^{(2)}$, given the super-sample. Theorem 3.4 provides a generalization bound in terms of the disintegrated mutual information between each individual element of $U^{(2)}$ and the output of the learning algorithm, $W$. The bound in Theorem 3.4 is an analogue of [6, Prop. 1] for Theorem 1.3. In this section as in Steinke and Zakynthinou [21], we only consider $\tilde{Z}^{(k)}$ and $U^{(k)}$ with $k = 2$, so we will drop the superscript from $U^{(k)}$. Let $U = (U_1, \dots, U_n)$. The proofs for the results of this section appear in Appendix D.

**Theorem 3.1.** *Fix $m \in [n]$ and let $J = (J_1, \dots, J_m)$ be a random subset of $[n]$, distributed uniformly among all subsets of size $m$ and independent from $W$, $\tilde{Z}^{(2)}$, and $U$. Then*

$$\text{EGE}_{\mathcal{D}}(\mathcal{A}) \leq \mathbb{E}\sqrt{\frac{2I^{\tilde{Z}^{(2)}}(W;U_J|J)}{m}}. \tag{2}$$

By applying Jensen's inequality to Theorem 3.1, we obtain

$$\text{EGE}_{\mathcal{D}}(\mathcal{A}) \leq \sqrt{\frac{2I(W;U_J|\tilde{Z}^{(2)},J)}{m}}. \tag{3}$$

Our next results in Theorem 3.2 let us compare Eq. (3) for different values of $m = |J|$.

**Theorem 3.2.** *Let $m_1 < m_2 \in [n]$, and let $J^{(m_1)}, J^{(m_2)}$ be random subsets of $[n]$, distributed uniformly among all subsets of size $m_1$ and $m_2$, respectively, and independent from $W$, $\tilde{Z}^{(2)}$, and $U$. Then*

$$\frac{I(W;U_{J^{(m_1)}}|\tilde{Z}^{(2)},J^{(m_1)})}{m_1} \leq \frac{I(W;U_{J^{(m_2)}}|\tilde{Z}^{(2)},J^{(m_2)})}{m_2}. \tag{4}$$

*Consequently, taking $m_2 = n$, for all $1 \leq m_1 \leq n$*

$$\mathbb{E}\sqrt{\frac{2\,I^{\tilde{Z}^{(2)}}(W;U_{J^{(m_1)}}|J^{(m_1)})}{m_1}} \leq \sqrt{\frac{2I(W;U|\tilde{Z}^{(2)})}{n}}. \tag{5}$$

**Corollary 3.3.** $\text{EGE}_{\mathcal{D}}(\mathcal{A}) \leq \sqrt{2\,I(W;U_J|\tilde{Z}^{(2)},J)/m}$. *The case $m = |J| = n$ is equivalent to Theorem 1.3. The bound is increasing in $m \in [n]$, and, the tightest bound is achieved when $m = |J| = 1$. Also, Eq. (5) shows our bound in Theorem 3.1 is tighter than Theorem 1.3 for $k = 2$.*

To further tighten Theorem 3.2 when $m = 1$, we show that we can pull the expectation over both $\tilde{Z}^{(2)}$ and $J$ outside the concave square-root function.

**Theorem 3.4.** *Let $J \sim \text{Unif}([n])$ (i.e., $m = 1$ above) be independent from $W$, $\tilde{Z}^{(2)}$, and $U$. Then*

$$\text{EGE}_{\mathcal{D}}(\mathcal{A}) \leq \mathbb{E}\sqrt{2I^{\tilde{Z}^{(2)},J}(W;U_J)} = \frac{1}{n}\sum_{i=1}^{n}\mathbb{E}\sqrt{2I^{\tilde{Z}^{(2)}}(W;U_i)}. \tag{6}$$

*Remark* 3.5. Theorem 3.4 is tighter than Theorem 1.3 since

$$\frac{1}{n}\sum_{i=1}^{n}\mathbb{E}\sqrt{2I^{\tilde{Z}^{(2)}}(W;U_i)} \leq \sqrt{\sum_{i=1}^{n}\frac{2}{n}I(W;U_i|\tilde{Z}^{(2)})} \leq \sqrt{\frac{2}{n}I(W;U|\tilde{Z}^{(2)})} \tag{7}$$

The first inequality is Jensen's, while the second follows from the independence of indices $U_i$.    ◁

## 3.1 Controlling CMI bounds using KL Divergence

It is often difficult to compute MI directly. One standard approach in the literature is to bound MI by the expectation of the KL divergence of the conditional distribution of the parameters given the data (the "posterior") with respect to a "prior". The statement below is adapted from Negrea et al. [15].

**Lemma 3.6.** *Let $X$, $Y$, and $Z$ be random elements. For all $\sigma(Z)$-measurable random probability measures $P$ on the space of $Y$,*

$$I^Z(X;Y) \leq \mathbb{E}^Z[\mathrm{KL}(\mathbb{P}^{X,Z}[Y] \,\|\, P)] \text{ a.s.,} \qquad \text{with a.s. equality for } P = \mathbb{E}^Z[\mathbb{P}^{X,Z}[Y]] = \mathbb{P}^Z[Y].$$

We refer to the conditional law of $W$ given $S$ as the *"posterior" of $W$ given $S$*, which we denote $Q = \mathbb{P}^S[W] = \mathbb{P}^{\tilde{Z}^{(2)},U}[W]$, and to $P$ as the *prior*. This can be used in combination with, for example, Lemma 3.6 and Theorem 1.3 to obtain that for any $\tilde{Z}^{(2)}$-measurable random prior $P(\tilde{Z}^{(2)})$

$$\mathrm{EGE}_{\mathcal{D}}(\mathcal{A}) \leq \sqrt{\frac{2\,I(W;U|\tilde{Z}^{(2)})}{n}} \leq \sqrt{\frac{2\mathbb{E}[\mathrm{KL}(Q \,\|\, P(\tilde{Z}^{(2)}))]}{n}}. \tag{8}$$

Note that the prior only has access to $\tilde{Z}^{(2)}$, therefore from its perspective the training set can take $2^n$ different values. Alternatively, combining Lemma 3.6 and Theorem 3.1 yields

$$\mathrm{EGE}_{\mathcal{D}}(\mathcal{A}) \leq \mathbb{E}\sqrt{\frac{2\,\mathbb{E}^{\tilde{Z}^{(2)}}I^{\tilde{Z}^{(2)}}(W;U_J|U_{J^c},J)}{m}} \leq \mathbb{E}\sqrt{\frac{2\mathbb{E}^{\tilde{Z}^{(2)}}[\mathrm{KL}(Q \,\|\, P(\tilde{Z}^{(2)},U_{J^c},J))]}{m}}. \tag{9}$$

In Eq. (9) the prior has access to $n - m$ samples in the training set, $S_{J^c}$, because $\tilde{Z}^{(2)}_{U_{J^c}} = S_{J^c}$. However, since $\tilde{Z}^{(2)}$ is known to the prior, the training set can take only $2^m$ distinct values from the point of view of the prior in Eq. (9). This is a significant reduction in the amount of information that can be carried by the indexes in $U_J$ about the output hypothesis. Consequently, priors can be designed to better exploit the dependence of the output hypothesis and the index set.

## 3.2 Tighter Generalization bound for the case $m = 1$

Since the strategy above controls MI-based expressions via KL divergences, one may ask whether a bound derived with similar tools, but directly in terms of KL, can be tighter than the combination Lemma 3.6 and Theorem 3.1. The following result shows that for $m = 1$ a tighter bound can be derived by pulling the expectation over both $U_{J^c}$ and $J$ outside the concave square-root function.

**Theorem 3.7.** *Let $J \sim Unif([n])$ be independent from $W$, $U$, and $\tilde{Z}^{(2)}$. Let $Q = \mathbb{P}^{\tilde{Z}^{(2)},U}[W]$ and $P$ be a $\sigma(\tilde{Z}^{(2)}, U_{J^c}, J)$-measurable random probability measure. Then*

$$\mathrm{EGE}_{\mathcal{D}}(\mathcal{A}) \leq \mathbb{E}\sqrt{2\,\mathrm{KL}(Q \,\|\, P)}. \tag{10}$$

Here, the KL divergence is between two $\sigma(\tilde{Z}^{(2)}, J, U)$-measurable random measures, so is random.

# 4 Generalization bounds for noisy, iterative algorithms

We apply this new class of generalization bounds to non-convex learning. We analyze the Langevin dynamics (LD) algorithm [8], following the analysis pioneered by Pensia et al. [16]. The example we set here is a blueprint for building bounds for other iterative algorithms. Our approach is similar to the recent advances by Li et al. [14] and Negrea et al. [15], employing data-dependent estimates to obtain easily simulated bounds. We find our new results allow us to exploit past iterates to obtain tighter bounds. The influence of past iterates is seen to take the form of a hypothesis test.

## 4.1 Bounding Generalization Error via Hypothesis Testing

The chain rule for KL divergence is a key ingredient of information-theoretic generalization error bounds for iterative algorithms [6, 14, 15, 16]. $\mathcal{W}^{\{0,\dots,T\}}$ denotes the space of parameters generated by an iterative algorithm in $T$ iterations. For any measure, $\nu$, on $\mathcal{W}^{\{0,\dots,T\}}$, and $W \sim \nu$, let $\nu_0$ denote the marginal law of $W_0$, and $\nu_{t|}$ denote the conditional law of $W_t$ given $W_0 \dots W_{t-1}$.

**Lemma 4.1** (Chain Rule for KL). *Let $Q, P$ be probability measures on $\mathcal{W}^{\{0,\dots,T\}}$ with $Q_0 = P_0$. The following lemma bounds the KL divergence involving the posterior for the terminal parameter with one involving the sum of the KL divergences over each individual step of the trajectory. Then*

$$\mathrm{KL}(Q_T \,\|\, P_T) \leq \mathrm{KL}(Q \,\|\, P) = \textstyle\sum_{t=1}^{T} Q_{0:(t-1)}[\mathrm{KL}(Q_{t|} \,\|\, P_{t|})]$$

The benefit of using the chain rule to analyze the iterative algorithm are two-fold: first, we gain analytical tractability; many bounds that appear in the literature implicitly require this form of incrementation [6, 14, 15, 16]. Second, and novel to the present work, the *information in the optimization trajectory can be exploited* to identify $U$ from the history of $W$.

In order to understand how the prior may take advantage of information from the optimization trajectory, consider applying Lemma 4.1 to the KL term in Eq. (9). We have

$$\mathrm{KL}(Q_T \| P_T(\tilde{Z}^{(2)}, U_{J^c}, J)) \leq \sum_{t=1}^{T} \mathbb{E}^{\tilde{Z}^{(2)}, U_{J^c}, J}[\mathrm{KL}(Q_{t|} \| P_{t|}(\tilde{Z}^{(2)}, U_{J^c}, J))].$$

Here $P_{t|}(\tilde{Z}^{(2)}, U_{J^c}, J)$ is a $\sigma(\tilde{Z}^{(2)}, U_{J^c}, J, W_{0:t-1})$-measurable random probability measure. The prior may use $U_{J^c}, \tilde{Z}^{(2)}$, and $J$ to reduce the number of possible values that $U$ can take to $2^{|J|}$. Moreover, since $U_J$ is constant during optimization, $W_0, W_1, W_2, \dots W_{t-1}$ may leak some information about $U_J$, and the prior can use this information to tighten the bound by choosing a $P_{t|}$ that achieves small $\mathrm{KL}(Q_{t|} \| P_{t|})$. In the special case where the prior can perfectly estimate $U_J$ from $W_0, W_1, W_2, \dots W_{t-1}$, we can set $P_{t|} = Q_{t|}$ and $\mathrm{KL}(Q_{t|} \| P_{t|})$ will be zero. As will be seen in the next subsection, we can explicitly design a prior that uses the information in the optimization trajectory for the LD algorithm.

The process by which the prior can learn from the trajectory can be viewed as an *online hypothesis test*, or binary decision problem, where the prior at time $t$ allocates belief between $2^m$ possible explanations, given by the possible values of $U_J$, based on the evidence provided by $W_0, \dots W_t$. If the prior is able to identify $U_J$ based on the $W$s then the bound stops accumulating, even if the gradients taken by subsequent training steps are large. This means that penalties for information obtained later in training are *discounted* based on the information obtained earlier in training.

### 4.2 Example: Langevin Dynamics Algorithm for Non-Convex Learning

We apply these results to obtain generalization bounds for a gradient-based iterative noisy algorithm, the Langevin Dynamics (LD) algorithm. For classification with continuous parameters, the 0-1 loss does not provide useful gradients. Typically we optimize a surrogate objective, based on a *surrogate loss*, such as cross entropy. Write $\tilde{\ell} : \mathcal{Z} \times \mathcal{W} \to \mathbb{R}$ for the surrogate loss and let $\tilde{R}_S(w) = \frac{1}{n} \sum_{i=1}^{n} \tilde{\ell}(Z_i, w)$ be the empirical surrogate risk. Let $\eta_t$ be the learning rate at time $t$, $\beta_t$ the inverse temperature at time $t$ and let $\varepsilon_t$ be sampled i.i.d. from $\mathcal{N}(0, \mathbb{I}_d)$. Then the LD algorithm iterates are given by

$$W_{t+1} = W_t - \eta_t \nabla \tilde{R}_S(W_t) + \sqrt{\frac{2\eta_t}{\beta_t}} \varepsilon_t. \tag{11}$$

**The prior** We will take $m = 1$, and construct a bespoke $\sigma(\tilde{Z}^{(2)}, U_{J^c}, J)$-measurable prior for this problem in order to apply Theorem 3.7. The prior is based on a *decision function*, $\theta : \mathbb{R} \to [0, 1]$, which at each time $t + 1$ takes in a $\sigma(W_0 \dots W_t)$-measurable *test statistic*, $\Delta Y_t$, and returns a *degree of belief* in favor of the hypothesis $U_J = 1$ over $U_J = 2$. The prior predicts an LD step by replacing the unknown (to the prior) contribution to the gradient of the data point at index $J$ with a $\hat{\theta}_t = \theta(\Delta Y_t)$-weighted average of the gradients due to each candidate $\{Z_{1,J}, Z_{2,J}\}$. The conditional law of the $t$th iterate under the prior is a $\sigma(\tilde{Z}^{(2)}, U_{J^c}, J, W_0, \dots W_t)$-measurable random measure, as required. The exact value of the test statistic is $\Delta Y_t = Y_{t,2} - Y_{t,1}$, here the $Y_{0,1} = Y_{0,2} = 0$ and $Y_{t,u}$ are defined by the formula in Eq. (13). The conditional law of the $t$th iterate under the prior is described by

$$W_{t+1} = W_t - \frac{\eta_t}{n} \left( \sum_{\substack{i=1 \\ i \neq J}}^{n} \nabla \tilde{\ell}(Z_i, W_t) + \hat{\theta}_t \nabla \tilde{\ell}(Z_{1,J}, W_t) + (1 - \hat{\theta}_t) \nabla \tilde{\ell}(Z_{2,J}, W_t) \right) + \sqrt{\frac{2\eta_t}{\beta_t}} \varepsilon_t. \tag{12}$$

The test statistic chosen is based on the log-likelihood-ratio test statistic for the independent mean 0 Gaussian random vectors $(\varepsilon_s)_{s=1}^{t}$, which is well known to be *uniformly most powerful* for the binary discrimination of means. Natural choices for $\theta$ are symmetric CDFs, since they treat possible values of $U$ symmetrically, and are monotone in the test statistic.

We define the *two-sample incoherence* at time $t$ by $\zeta_t = \nabla \tilde{\ell}(Z_{1,J}, W_t) - \nabla \tilde{\ell}(Z_{2,J}, W_t)$. $\Theta$ denotes the set of measurable $\theta : \mathbb{R} \to [0, 1]$. $Y_{0,1} = Y_{0,2} = 0$, and for $t \geq 1$, $Y_{t,1}$ and $Y_{t,2}$ are given by (for $u \in \{1, 2\}$)

$$Y_{t,u} \triangleq \sum_{i=1}^{t} \frac{\beta_{i-1}}{4\eta_{i-1}} \left\| W_i - W_{i-1} + \eta_{i-1} \frac{n-1}{n} \nabla \tilde{R}_{S_{J^c}}(W_{i-1}) + \frac{\eta_{i-1}}{n} \nabla \tilde{\ell}(Z_{u,J}, W_{i-1}) \right\|^2. \tag{13}$$

**Theorem 4.2** (Generalization bound for LD algorithm). *Let $\{W_t\}_{t\in[T]}$ denote the iterates of the LD algorithm. If $\ell(Z,w)$ is $[0,1]$-bounded then*

$$\mathbb{E}\left[R_{\mathcal{D}}(W_T)-\hat{R}_S(W_T)\right] \leq \frac{1}{n\sqrt{2}}\inf_{\theta\in\Theta}\mathbb{E}\sqrt{\sum_{t=0}^{T-1}\mathbb{E}^{\tilde{Z}^{(2)},U,J}\beta_t\eta_t\|\zeta_t\|^2\left(\mathbb{1}\{U_J=1\}-\theta\left(Y_{t,2}-Y_{t,1}\right)\right)^2}.$$

(14)

*Remark* 4.3. For $\theta\in\Theta$ with $1-\theta(x)=\theta(-x)$, Eq. (14) simplifies to

$$\mathbb{E}\left[R_{\mathcal{D}}(W_T)-\hat{R}_S(W_T)\right] \leq \frac{1}{n\sqrt{2}}\mathbb{E}\sqrt{\sum_{t=0}^{T-1}\mathbb{E}^{\tilde{Z}^{(2)},U,J}\beta_t\eta_t\|\zeta_t\|^2\theta^2\left(-1^{U_J}\left(Y_{t,2}-Y_{t,1}\right)\right)}.$$

(15)

For instance $\theta(x)=\frac{1}{2}+\frac{1}{2}\tanh(x)$ and $\theta(x)=\frac{1}{2}+\frac{1}{2}\mathrm{sign}(x)$ satisfy $1-\theta(x)=\theta(-x)$. ◁

*Remark* 4.4. By the law of total expectation, for any $\theta\in\Theta$, $\mathrm{EGE}_{\mathcal{D}}(\mathcal{A}) \leq \frac{1}{2\sqrt{2}n}\mathbb{E}[V_1+V_2]$, where

$$V_u \triangleq \sqrt{\sum_{t=0}^{T-1}\mathbb{E}^{\tilde{Z}^{(2)},U_{J^c},J,U_J=u}\beta_t\eta_t\|\zeta_t\|^2\left(\mathbb{1}\{u=1\}-\theta\left(Y_{t,2}-Y_{t,1}\right)\right)^2}, \quad u\in\{1,2\}.$$

(16)

To estimate $V_u$ ($u\in\{1,2\}$) for fixed $J$, the training set is $S_u=\{Z_1,\dots,Z_{J-1},\tilde{Z}_{u,J},Z_{J+1},\dots,Z_n\}$. Hence $V_1,V_2$ can be simulated from just $n+1$ data points $\left(Z_1,\dots,Z_{J-1},Z_{J+1},\dots,Z_n,\tilde{Z}_{1,J},\tilde{Z}_{2,J}\right) \sim \mathcal{D}^{\otimes(n+1)}$. ◁

The generalization bound in Eq. (14) does not place any restrictions on the learning rate or Lipschitz continuity of the loss or its gradient. In the next corollary we study the asymptotic properties of the bound in Eq. (14) when $\tilde{\ell}$ is $L$-Lipschitz. Then, we draw a comparison between the bound in this paper and some of the existing bounds in the literature.

**Corollary 4.5.** *Under the assumption that $\tilde{\ell}$ is $L$-Lipschitz, we have $\|\zeta_t\| \leq 2L$. Then, the generalization bound in Eq.* (14) *can be upper-bounded as*

$$\mathbb{E}(R_{\mathcal{D}}(W_T)-R_S(W_T)) \leq \frac{\sqrt{2}L}{n}\inf_{\theta\in\Theta}\mathbb{E}\sqrt{\sum_{t=0}^{T-1}\mathbb{E}^{\tilde{Z}^{(2)},U,J}\beta_t\eta_t\left(\mathbb{1}\{U_J=1\}-\theta\left(Y_{t,2}-Y_{t,1}\right)\right)^2}.$$

(17)

*Remark* 4.6. Under an $L$-Lipschitz assumption, for the LD algorithm, Li et al. [14, Thm. 9] have

$$\mathbb{E}\left[R_{\mathcal{D}}(W_T)-R_S(W_T)\right] \leq \frac{\sqrt{2}L}{n}\sqrt{\sum_{t=0}^{T-1}\beta_t\eta_t}.$$

(18)

We immediately see that Eq. (17) provides a constant factor improvement over Eq. (18) by naïvely using $\theta:x\mapsto 1/2$. Our bound has order-wise improvement with respect to $n$ over that of Bu et al. [6] and Pensia et al. [16] under the $L$-Lipschitz assumption. Negrea et al. [15, App. E.1] obtain

$$\mathbb{E}\left[R_{\mathcal{D}}(W_T)-R_S(W_T)\right] \leq \frac{L}{2(n-1)}\sqrt{\sum_{t=0}^{T-1}\beta_t\eta_t}.$$

(19)

which is a constant factor better than our bound for the choice $\theta:x\mapsto 1/2$. However, this $\theta$ essentially corresponds to no hypothesis test, yielding the same prior as in [15]. For more sophisticated choices of decision function ($\theta$), even under a Lipschitz-surrogate loss assumption, it is difficult to compare our bound with related work because the exact impact of $\theta$-discounting is difficult to quantify analytically. ◁

*Remark* 4.7. A prevailing method for analyzing the generalization error in [6, 14, 15, 16] for iterative algorithms is via the chain rule for KL, using priors for the joint distribution of weight vectors that are Markov, i.e., given the $t$th weight, the $(t+1)$th weight is conditionally independent from the trajectory so far. Existing results using this approach accumulate a "penalty" for each step. In [6, 14, 15], the penalty terms are, respectively, the squared Lipschitz constant, the squared norm of the gradients, and the trace of the minibatch gradient covariance. The penalty term in our paper is the squared norm of "two-sample incoherence", defined in Theorem 4.2 as the squared norm of the difference between the gradient of a randomly selected training point and the held-out point. However, the use of chain rule along with existing "Markovian" priors introduces a source of looseness, i.e., the accumulating penalty may diverge to $+\infty$ yielding vacuous bounds (as seen in Fig. 1). *The distinguishing feature of our data-dependent CMI analysis is that the penalty terms get "filtered" by the online hypothesis test via our non-Markovian prior, i.e., our prediction for $t+1$ depends on whole trajectory.* When the true index can be inferred from the previous weights, then the penalty essentially stops accumulating. ◁

### 4.2.1 Empirical Results

In order to better understand the effect of discounting and the degree of improvement due to our new bounds and more sophisticated prior, we turn to simulation studies. We present and compare the empirical evaluations of the generalization bound in Theorem 4.2 with the data-dependent generalization bounds in Li et al. [14] and Negrea et al. [15]. For brevity, many of the details behind our implementation are deferred to Appendix G. The functional form of our bounds and [14, 15] are nearly identical as all of them use the chain rule for KL divergence. Nevertheless, the summands appearing in the bounds are different. The bound in [14] depends on the squared surrogate loss gradients norm, and the generalization bound in Negrea et al. [15] depends on the squared norm of *training set incoherence* defined as $\|\nabla \tilde{\ell}(Z_J, W_t) - \frac{1}{n-1} \sum_{i \in [n], i \neq J} \nabla \tilde{\ell}(Z_i, W_t)\|^2$ where the training set is $\{Z_1, \ldots, Z_n\}$ and $J \sim \text{Unif}([n])$. The first key difference between our bound and others is that the summand in our bound consists of two terms: squared norm of the two-sample incoherence, i.e., $\|\zeta_t\|^2$, and the squared error probability of a hypothesis test at time $t$, given by the term $\left( \mathbb{1}\{U_J = 1\} - \theta \left( \sum_{i=0}^{t} (Y_{i,2} - Y_{i,1}) \right) \right)^2$ in our bound. A consequence of this, and the second fundamental difference between our bound and existing bounds, is that our bound exhibits a trade-off in $\|\zeta_t\|^2$ because large $\|\zeta_t\|^2$ will make the error of the hypothesis test small on future iterations, whereas the bounds in [14, 15] are uniformly increasing with respect to the squared norm of surrogate loss gradients and the training set incoherence, respectively. In this section we empirically evaluate and compare our bound with related work across various neural network architectures and datasets.

Using Monte Carlo (MC) simulation, we compared estimates of our expected generalization error bounds with estimates of the bound from [14, 15] for the MNIST [13], CIFAR10 [12], and Fashion-MNIST [23] datasets in Fig. 1 and Table 1. For all the plots we consider $\theta(x) = \frac{1}{2}(1 + \text{erf}(x))$ for our bound. Also, in the last row of Table 1, we report the *unbiased estimate* of our bound optimized over the choice of $\theta$ function. We plot the squared norm of the two sample incoherence and training set incoherence, as well as the squared error probability of the hypothesis test. Fig. 1 and Table 1 show that our bound is tighter, and remain non-vacuous after many more iterations. We also observe that the variances for MC estimates of our bound are smaller than those of Negrea et al. [15], and it is also smaller than Li et al. [14] for CIFAR10 and MNIST-CNN experiments. Moreover, we observe that the error probability of the hypothesis test decays with the number of iterations, which matches the intuition that, as one observes more noisy increments of the process, one is more able to determine which point is contributing to the gradient. For CIFAR10, $\|\zeta_t\|^2$ is large because the generalization gap is large. However, as mentioned in the beginning of this section, large $\|\zeta_t\|^2$ makes the hypothesis testing easier on subsequent iterations. For instance, after iteration 600 the error is vanishingly small for CIFAR10 experiments which results in a plateau region in the bound. We can also observe the same phenomenon for the Fashion-MNIST experiment. This property distinguishes our bound from those in [14, 15].

Results for MNIST with CNN demonstrate that $\|\zeta_t\|^2$ and training set incoherence are close to each other. The reason behind this observations is that the generalization gap is small. Also, for this experiment the performance of the hypothesis testing is only slightly better than random guessing since the generalization gap is small, and it is difficult to distinguish the training samples from the test samples. This observation explains why our generalization bound is close to that of [15]. Nevertheless, the hypothesis testing performance improves with more training iterations, leading the two bounds to diverge, with our new bound performing better at later iterations. Finally, the scaling of our bound with respect to the number of iteration is tighter than in the bounds in [14, 15] as can be seen in Fig. 1.

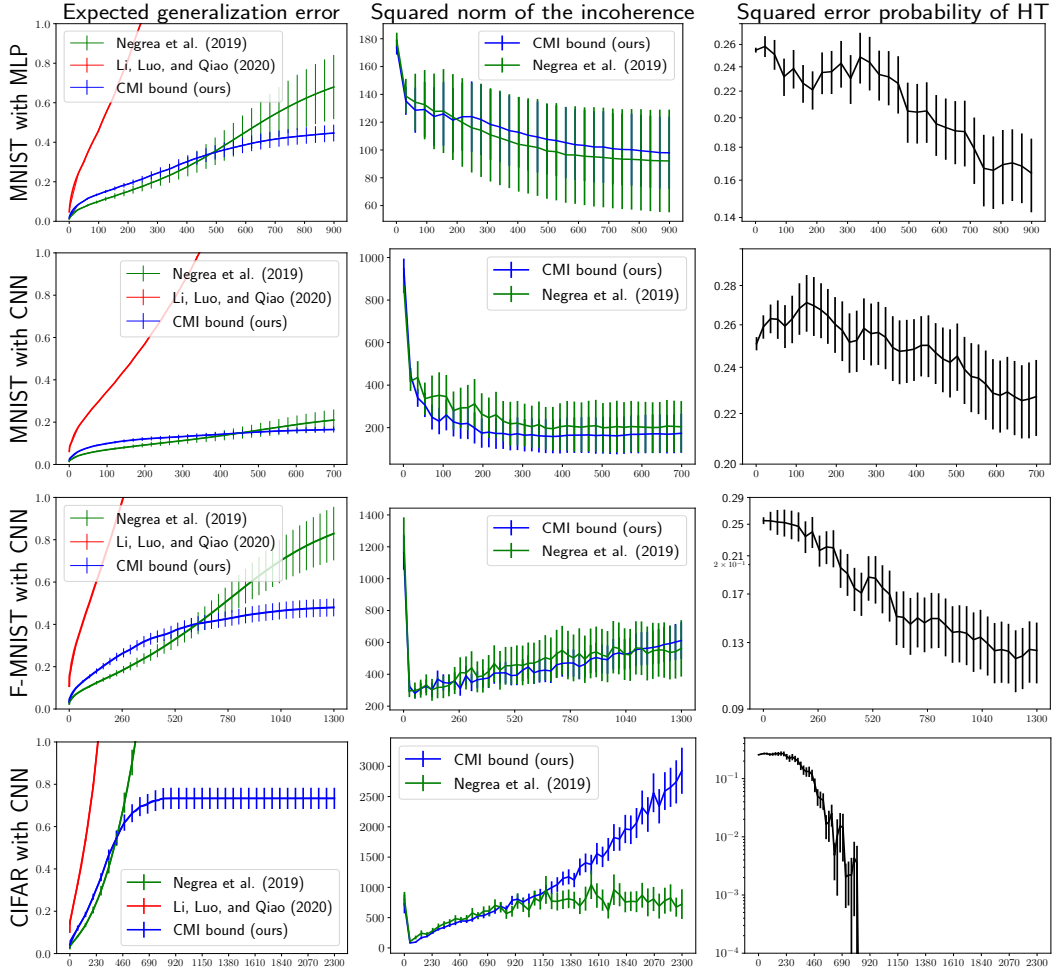

Figure 1: Numerical results for various datasets and architectures. All the x-axes represent the training iteration. The plots in the first column depict a Monte Carlo estimate of our bounds with that of Li et al. [14] and Negrea et al. [15]. The plots in the second column compare the mean of the *training set incoherence* in [15] with the two-sample incoherence in our bound. Finally, the plots in the third column show the mean of the squared error probability of the hypothesis testing performed by the proposed prior in our bound.

| | MNIST-MLP | MNIST-CNN | CIFAR10-CNN | FMNIST-CNN |
|---|---|---|---|---|
| Training error | $4.33 \pm 0.01\%$ | $2.59 \pm 0.01\%$ | $9.39 \pm 0.36\%$ | $7.96 \pm 0.03\%$ |
| Generalization error | $0.88 \pm 0.01\%$ | $0.55 \pm 0.01\%$ | $32.89 \pm 0.44\%$ | $3.71 \pm 0.03\%$ |
| Negrea et al. [15] | $67.93 \pm 16.25\%$ | $20.98 \pm 5.01\%$ | $4112.63 \pm 567.08\%$ | $82.89 \pm 12.64\%$ |
| Li et al. [14] | $600.29 \pm 1.99\%$ | $245.03 \pm 2.37\%$ | $20754.32 \pm 75.95\%$ | $598.62 \pm 3.21\%$ |
| **CMI (Ours)** | $\mathbf{44.65 \pm 4.27\%}$ | $\mathbf{16.51 \pm 1.41\%}$ | $\mathbf{71.76 \pm 4.82\%}$ | $\mathbf{48.01 \pm 4.22\%}$ |
| **CMI-OPT(Ours)** | $\mathbf{39.06 \pm 5.52\%}$ | $\mathbf{13.24 \pm 1.53\%}$ | $\mathbf{63.00 \pm 5.97\%}$ | $\mathbf{41.17 \pm 5.85\%}$ |

Table 1: Summary of the results. The generalization bounds are reported at the end of training.

**Acknowledgments**

The authors would like to thank Blair Bilodeau and Yasaman Mahdaviyeh for feedback on drafts of this work, and Shiva Ketabi for helpful discussions on the implementation of the bounds.

**Funding**

MH is supported by the Vector Institute. JN is supported by an NSERC Vanier Canada Graduate Scholarship, and by the Vector Institute. DMR is supported by an NSERC Discovery Grant and an Ontario Early Researcher Award. This research was carried out in part while MH, JN, GKD, and DMR were visiting the Institute for Advanced Study. JN's visit to the Institute was funded by an NSERC Michael Smith Foreign Study Supplement. Resources used in preparing this research were provided, in part, by the Province of Ontario, the Government of Canada through CIFAR, and companies sponsoring the Vector Institute `www.vectorinstitute.ai/partners`.

**Broader Impact**    This work builds upon the community's understanding of generalization error for machine learning methods. This has a positive impact on the scientific advancement of the field, and may lead to further improvements in our understanding, methodologies and applications of machine learning and AI. While there are not obvious direct societal implications of the present work, the indirect and longer term impact on society may be positive, negative or both depending on how, where and for what machine learning method that will have benefited from our research are used in the future.

## Footnotes

[1] Letting $\phi$ satisfy $\phi(Z) = I^Z(X;Y)$ a.s., define $I(X,Y|Z = z) = \phi(z)$. This notation is necessarily well defined only up to a null set under the marginal distribution of $Z$.

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
