[Supplementary Material · cmistep_neurips.pdf]

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

# A  CMI, Membership Attack, and Fano's Inequality

Let $\tilde{Z}^{(k)}$, $U^{(k)}$, and $S$ as in Definition 1.2. Consider the following hypothesis testing problem. Assume a decision maker observes $W$ and wishes to recover $U^{(k)}$ by having access to the super-sample $\tilde{Z}^{(k)}$. For any estimate $\hat{U} = \Psi(W, \tilde{Z}^{(k)})$, we have the Markov chain

$$U^{(k)} \to S \to W \to \widehat{U^{(k)}}.$$

and so, combined with the fact that $U^{(k)}$ is uniformly distributed over a set of size $k^n$, we can invoke Fano's inequality to bound the error probability of the decision maker. In particular,

$$\inf_{\Psi} \mathbb{P}\left[ \Psi\left(W, \tilde{Z}^{(k)}\right) \neq U^{(k)} \right] \geq 1 - \frac{I(W; U^{(k)}|\tilde{Z}^{(k)}) + \log 2}{n \log k}.$$

Hence, $I(W; U^{(k)}|\tilde{Z}^{(k)})$ provides a lower bound on the hardness of the hypothesis testing problem, where one wants to identify the training sample given access to $\tilde{Z}^{(k)}$ and $W$.

Some interpretation of our result is helpful. Consider an adversary who has access to the supersample $\tilde{Z}^{(k)}$ and wishes to identify the training set that was used for the training after observing the output of a learning algorithm $W$. Our result here showed that the CMI upperbounds the success probability of *every* adversary. Also, recall that the CMI upper bounds the expected generalization error. In the literature of data privacy in machine learning, this problem is known as Membership Attack [20], and it is empirically observed that a machine learning model leaks information about its training set when the generalization error is large [20]. Our result in this section provides a formal connection between generalization and this specific membership attack problem.

# B  Matching the leading coefficient of Theorem 1.1 with $\mathrm{CMI}_{\mathcal{D}}^{k}(\mathcal{A})$

**Theorem B.1.** *Let $\mathrm{CMI}_{\mathcal{D}}^{k}(\mathcal{A})$ as defined in the introduction. Then, for $k > 2$*

$$\mathrm{EGE}_{\mathcal{D}}(\mathcal{A}) \leq \frac{\mathrm{CMI}_{\mathcal{D}}^{k}(\mathcal{A})}{\lambda^\star} + \frac{\exp(\frac{\lambda^\star}{n}) - \frac{\lambda^\star}{n} - 1}{\frac{\lambda^\star}{n}} \frac{k^3 + 7k^2 - 8k - 16}{4(k^3 - 2k^2)},$$

*where*

$$\lambda^\star = n\mathfrak{W}\left( \left(\frac{4(k^3 - 2k^2)\mathrm{CMI}_{\mathcal{D}}^{k}(\mathcal{A})}{n(k^3 + 7k^2 - 8k - 16)} - 1\right)\exp(-1)\right) + n,$$

*and $\mathfrak{W}$ is the Lambert W function.*

The proof is deferred to Appendix C. Here, we quantitatively compare Theorem 1.1, Theorem 1.3, and Theorem B.1, we consider the case that the output of $\mathcal{A}$ takes value in a finite set and $k \to \infty$. In this case, Theorem B.1 can rephrased as

$$\mathrm{EGE}_{\mathcal{D}}(\mathcal{A}) \leq \lim_{k \to \infty} \frac{\mathrm{CMI}_{\mathcal{D}}^{k}(\mathcal{A})}{\lambda^\star} + \frac{\exp(\frac{\lambda^\star}{n}) - \frac{\lambda^\star}{n} - 1}{\frac{\lambda^\star}{n}} \frac{k^3 + 7k^2 - 8k - 16}{4(k^3 - 2k^2)}$$

$$= \frac{\mathrm{IOMI}_{\mathcal{D}}(\mathcal{A})}{\lambda^\star_\infty} + \frac{\exp(\frac{\lambda^\star_\infty}{n}) - \frac{\lambda^\star_\infty}{n} - 1}{4\frac{\lambda^\star_\infty}{n}} \tag{20}$$

where $\lambda^\star_\infty = n\mathfrak{W}\left(\left(\frac{4\mathrm{IOMI}_{\mathcal{D}}(\mathcal{A})}{n} - 1\right)\exp(-1)\right) + n$. In the next plot, we compare the values of the bounds in Theorem 1.1, Theorem 1.3, and Theorem B.1 assuming $\mathrm{IOMI}_{\mathcal{D}}(\mathcal{A}) = 1$. As seen, the bound in Theorem B.1 provides much tighter constant compared with Theorem 1.3, and it matches with Theorem 1.1.

Figure 2: Comparison between constants of Theorem 1.1, Theorem 1.3, and Theorem B.1 for the case $k \to \infty$.

## C  Proofs of Section 2

*Proof of Theorem 2.1.*  By the chain rule for the mutual information, we have

$$I(W; U^{(k)}, \tilde{Z}^{(k)}) = I(W; \tilde{Z}^{(k)}) + I(W; U^{(k)} | \tilde{Z}^{(k)}). \tag{21}$$

Since $S$ is $\sigma(\tilde{Z}^{(k)}, U^{(k)})$-measurable, $I(W; U^{(k)}, \tilde{Z}^{(k)}) = I(W; S, U^{(k)}, \tilde{Z}^{(k)})$. But then $W$ is independent of $\tilde{Z}^{(k)}, U^{(k)}$ given $S$, hence $I(W; S, U^{(k)}, \tilde{Z}^{(k)}) = I(W; S)$. The result follows from the nonnegativity of mutual information. $\qquad\square$

*Proof of Theorem 2.2.*  By Theorem 2.1, $I(W; U^{(k)} | \tilde{Z}^{(k)}) = I(W; S) - I(W; \tilde{Z}^{(k)})$. Therefore, in order to prove the claim, we need to show $\lim_{k \to \infty} I(W; \tilde{Z}^{(k)}) = 0$ when $I(W; S)$ is finite.

Recall that $\mathcal{A}$ is a probability kernel from the space of tuples in $\mathcal{Z}$ to $\mathcal{W}$. Assume $\mathcal{W} = \{w_1, \ldots, w_m\}$. For each $l \in [m]$, let $\kappa_l(S) = \mathbb{P}^S[W = w_l]$ and $f_l : \mathcal{Z}^{kn} \to [0, 1]$ be a measurable function defined as

$$f_l(\tilde{Z}^{(k)}) = \frac{1}{k^n} \sum_{u \in [k]^n} \kappa_l(\tilde{Z}_u^{(k)}).$$

Letting $z, z' \in \mathcal{Z}^{kn}$ be two super-samples that only differ in one element, it is straightforward to see that

$$|f_l(z) - f_l(z')| \le \frac{1}{k}.$$

Therefore, we can invoke McDiarmid's inequality to obtain

$$\mathbb{P}[|f_l(\tilde{Z}^{(k)}) - \mathbb{E}[f_l(\tilde{Z}^{(k)})]| \ge \varepsilon] \le \exp\left(-\frac{2k\varepsilon^2}{n}\right). \tag{22}$$

Also, we have $\mathbb{E}[f_l(\tilde{Z}^{(k)})] = \mathbb{P}[W = w_l]$ as each element of $\tilde{Z}^{(k)}$ is IID. Hence, $f_l(\tilde{Z}^{(k)}) \to \mathbb{P}[W = w_l]$ in probability as $k$ diverges.

By the definition of mutual information and KL divergence,

$$
\begin{aligned}
I(W; \tilde{Z}^{(k)}) &= \mathbb{E}[\mathrm{KL}(\mathbb{P}^{\tilde{Z}^{(k)}}[W] \,\|\, \mathbb{P}[W])] \\
&= \mathbb{E}\left[\mathrm{KL}\left(\frac{1}{k^n} \sum_{u \in [k]^n} \mathbb{P}^{\tilde{Z}_u^{(k)}}[W] \,\|\, \mathbb{P}[W]\right)\right] \\
&= \mathbb{E}\left[\sum_{l=1}^{m} \frac{1}{k^n} \sum_{u \in [k]^n} \kappa_l(\tilde{Z}_u^{(k)}) \log \frac{\frac{1}{k^n} \sum_{u \in [k]^n} \kappa_l(\tilde{Z}_u^{(k)})}{\mathbb{P}[W = w_l]}\right] \\
&= \sum_{l=1}^{m} \mathbb{E}\left[f_l(\tilde{Z}^{(k)}) \log \frac{f_l(\tilde{Z}^{(k)})}{\mathbb{P}[W = w_l]}\right]. \tag{23}
\end{aligned}
$$

Defining $\phi_l : [0,1] \to \mathbb{R}$ as $\phi_l(x) = x \log \frac{x}{\mathbb{P}[W=w_l]}$, we have established

$$I(W; \tilde{Z}^{(k)}) = \sum_{l=1}^{m} \mathbb{E}\big[\phi_l\big(f_l(\tilde{Z}^{(k)})\big)\big]. \tag{24}$$

Note that $\phi_l$ is a continuous and bounded function. By a standard result [7, Thm. 2.3.4], $f_l(\tilde{Z}^{(k)}) \to \mathbb{P}[W = w_l]$ in probability implies that

$$\mathbb{E}\big[\phi_l\big(f_l(\tilde{Z}^{(k)})\big)\big] \to \mathbb{E}\big[\phi_l\big(\mathbb{P}[W = w_l]\big)\big] = 0,$$

as $k$ goes to infinity. Using this, we conclude that $I(W; \tilde{Z}^{(k)}) \to 0$ as $k$ diverges, as was to be shown. $\qquad\square$

*Proof of Theorem B.1.* For any $k \in \mathbb{N}$ define

$$\rho_i^{(k)}(m) = \begin{cases} -1 & \text{if } m = i, \\ \frac{1}{k-1} & \text{otherwise} \end{cases}.$$

where $m$ and $i \in [k]$. Consider random variables $\tilde{Z}^{(k)}$, $U$, and $W$ as in the definition of $\mathrm{CMI}_{\mathcal{D}}^k(\mathcal{A})$. Also, let $\hat{U} \overset{d}{=} U$ and $\hat{U} \perp\!\!\!\perp (\tilde{Z}^{(k)}, W, U)$. Let $f : \mathcal{Z}^{kn} \times [k]^n \times \mathcal{W} \to [-1, 1]$ be given by

$$f(\tilde{z}^{(k)}, u, w) = \frac{1}{n} \sum_{j=1}^{n} \sum_{i=1}^{k} \rho_i^{(k)}(u_j) \ell(w, z_{i,j}).$$

Then, by the Donsker–Varadhan variational formula [5, Prop. 4.15] of the KL divergence we obtain

$$\begin{aligned} \mathrm{CMI}_{\mathcal{D}}^k(\mathcal{A}) &= \mathbb{E}[\mathrm{KL}(\mathbb{P}^{\tilde{Z}^{(k)}, W}[U] \,\|\, \mathbb{P}[U])] \\ &\geq \lambda \mathbb{E}[f(\tilde{Z}^{(k)}, U, W)] - \mathbb{E}\log \mathbb{E}^{\tilde{Z}^{(k)}, W}[\exp(\lambda f(\tilde{Z}^{(k)}, \hat{U}, W))] \end{aligned} \tag{25}$$

It is straightforward to see that the first term in the RHS of Eq. (25) is $\mathbb{E}[f(\tilde{Z}^{(k)}, U, W)] = \mathrm{EGE}_{\mathcal{D}}(\mathcal{A})$. In what follows we analyze the second term in the RHS of Eq. (25). We begin with

$$\begin{aligned} \mathbb{E}\log \mathbb{E}^{\tilde{Z}^{(k)}, W}[\exp(\lambda f(\tilde{Z}^{(k)}, \hat{U}, W))] &= \mathbb{E}\log \mathbb{E}^{\tilde{Z}^{(k)}, W}[\prod_{j=1}^{n} \exp(\frac{\lambda}{n} \sum_{i=1}^{k} \rho_i^{(k)}(\hat{U}_j) \ell(W, Z_{i,j})] \qquad (26) \\ &= \mathbb{E}\log \prod_{j=1}^{n} \mathbb{E}^{\tilde{Z}^{(k)}, W} \exp\big(\frac{\lambda}{n} \sum_{i=1}^{k} \rho_i^{(k)}(\hat{U}_j) \ell(W, Z_{i,j})\big) \\ &\leq \mathbb{E} \sum_{j=1}^{n} \big(\exp(\frac{\lambda}{n}) - \frac{\lambda}{n} - 1\big) \mathbb{E}^{\tilde{Z}^{(k)}, W}[\sum_{i=1}^{k} \rho_i^{(k)}(\hat{U}_j) \ell(W, Z_{i,j})]^2 \quad (27) \end{aligned}$$

The first step follows from the independence of the elements in $\hat{U}$. The last inequality is obtained by the Bennet's inequality [5, Thm. 2.9] on the moment generating function and the fact that the elements of $\hat{U}$ are independent of $(\tilde{Z}^{(k)}, W)$. Also, we have used $\mathbb{E}^{\tilde{Z}^{(k)}, W} \sum_{i=1}^{k} \rho_i^{(k)}(\hat{U}_j) \ell(W, Z_{i,j}) = 0$

since $\mathbb{E}\rho_i^{(k)}(\hat{U}_j) = 0$ and $|\sum_{i=1}^{k}\rho_i^{(k)}(\hat{U}_j)\ell(W,Z_{i,j})| \le 1$ a.s.. For a fixed $j$, from Eq. (27) we obtain

$$\mathbb{E}[\sum_{i=1}^{k}\rho_i^{(k)}(\hat{U}_j)\ell(W,Z_{i,j})]^2 = \frac{1}{k}\mathbb{E}\sum_{\tilde{i}=[k]}\left[\frac{1}{k-1}\sum_{i\in[k],i\neq\tilde{i}}\ell(W,Z_{i,j}) - \ell(W,Z_{\tilde{i},j})\right]^2 \tag{28}$$

$$= \frac{1}{k^2}\mathbb{E}\left[\sum_{u_j\in[k]}\mathbb{E}^{\tilde{Z}^{(k)},U_j=u_j}\sum_{\tilde{i}=[k]}\left[\frac{1}{k-1}\sum_{i\in[k],i\neq\tilde{i}}\ell(W,Z_{i,j}) - \ell(W,Z_{\tilde{i},j})\right]^2\right] \tag{29}$$

$$= \frac{1}{k^2}\mathbb{E}\left[\sum_{u_j\in[k]}\mathbb{E}^{\tilde{Z}^{(k)},U_j=u_j}\left[\sum_{\tilde{i}=[k],\tilde{i}\neq u_j}\left[\frac{1}{k-1}\ell(W,Z_{u_j,j}) + \frac{1}{k-1}\sum_{i\in[k],i\neq\{\tilde{i},u_j\}}\ell(W,Z_{i,j}) - \ell(W,Z_{\tilde{i},j})\right]^2\right.\right.$$
$$\left.\left.+ \left[\frac{1}{k-1}\sum_{i\in[k],i\neq\tilde{i}}\ell(W,Z_{i,j}) - \ell(W,Z_{u_j,j})\right]^2\right]\right] \tag{30}$$

$$\le \frac{1}{k^2}\mathbb{E}\left[\sum_{u_j\in[k]}\mathbb{E}^{\tilde{Z}^{(k)},U_j=u_j}\left[\sum_{\tilde{i}=[k],\tilde{i}\neq u_j}\left[\left(\frac{1}{k-1} + \frac{1}{k-1}\sum_{i\in[k],i\neq\{\tilde{i},u_j\}}\ell(W,Z_{i,j})\right)^2 + \ell(W,Z_{\tilde{i},j})^2\right.\right.\right.$$
$$\left.\left.\left.- 2\ell(W,Z_{\tilde{i},j})\frac{1}{k-1}\sum_{i\in[k],i\neq\{\tilde{i},u_j\}}\ell(W,Z_{i,j})\right] + \left[\left[\frac{1}{k-1}\sum_{i\in[k],i\neq\tilde{i}}\ell(W,Z_{i,j})\right]^2 + 1\right]\right]\right] \tag{31}$$

$$= \frac{1}{k^2}\sum_{u_j\in[k]}\mathbb{E}\left[\sum_{\tilde{i}\in[k],u_j\neq\tilde{i}}\mathbb{E}^{W}\left[\left(\frac{1}{k-1} + \frac{1}{k-1}\sum_{i\in[k],i\neq\{\tilde{i},u_j\}}\ell(W,Z_{i,j})\right)^2 + \ell(W,Z_{\tilde{i},j})^2\right.\right.$$
$$\left.\left.- 2\ell(W,Z_{\tilde{i},j})\frac{1}{k-1}\sum_{i\in[k],i\neq\{\tilde{i},u_j\}}\ell(W,Z_{i,j})\right] + \mathbb{E}^{W}\left[\left[\frac{1}{k-1}\sum_{i\in[k],i\neq\tilde{i}}\ell(W,Z_{i,j})\right]^2 + 1\right]\right]. \tag{32}$$

Here, Eq. (28) is obtained by taking the expectation over $\hat{U}_j$, the definition of $\rho_i^{(k)}(\hat{U}_j)$, and $\hat{U} \perp\!\!\!\perp (\tilde{Z}^{(k)}, W)$. Then, Eq. (29) is by the law of iterated expectations. Specifically, we condition on $U_j$, and recall that based on the Definition 1.2 the $j$-th training sample is $Z_{U_j,j}$. Step Eq. (30) is by some manipulations. Eq. (31) is obtained by

$$\left[\frac{1}{k-1}\ell(W,Z_{u_j,j}) + \frac{1}{k-1}\sum_{i\in[k],i\neq\{\tilde{i},u_j\}}\ell(W,Z_{i,j}) - \ell(W,Z_{\tilde{i},j})\right]^2 \le \left(\frac{1}{k-1} + \frac{1}{k-1}\sum_{i\in[k],i\neq\{\tilde{i},u_j\}}\ell(W,Z_{i,j})\right)^2$$
$$+ \ell(W,Z_{\tilde{i},j})^2 - \ell(W,Z_{\tilde{i},j})\frac{2}{k-1}\sum_{i\in[k],i\neq\{\tilde{i},u_j\}}\ell(W,Z_{i,j}),$$

and

$$\left[\frac{1}{k-1}\sum_{i\in[k],i\neq\tilde{i}}\ell(W,Z_{i,j}) - \ell(W,Z_{u_j,j})\right]^2 \le 1 + \left(\frac{1}{k-1}\sum_{i\in[k],i\neq\tilde{i}=u_j}\ell(W,Z_{i,j})\right)^2.$$

Finally last step is obtained by changing the order of the expectation over $W$ and $\tilde{Z}^{(k)}$. Then, we can simplify Eq. (32) by considering

$$\mathbb{E}^{W}\left[\left(\frac{1}{k-1} + \frac{1}{k-1}\sum_{i\in[k],i\neq\{\tilde{i},u_j\}}\ell(W,Z_{i,j})\right)^2 + \ell(W,Z_{\tilde{i},j})^2 - 2\ell(W,Z_{\tilde{i},j})\frac{1}{k-1}\sum_{i\in[k],i\neq\{\tilde{i},u_j\}}\ell(W,Z_{i,j})\right]$$

$$\le R_{\mathcal{D}}(W)^2\frac{-k^2+k+2}{(k-1)^2} + R_{\mathcal{D}}(W)\frac{k^2+k-5}{(k-1)^2} + \frac{1}{(k-1)^2} \triangleq A_1(k,R_{\mathcal{D}}(W)) \tag{33}$$

$$\mathbb{E}^{W}\left[\left[\frac{1}{k-1}\sum_{i\in[k],i\neq\tilde{i}}\ell(W,Z_{i,j})\right]^2 + 1\right] \le R_{\mathcal{D}}(W)^2\frac{k-2}{k-1} + R_{\mathcal{D}}(W)\frac{1}{k-1} + 1 \triangleq A_2(k,R_{\mathcal{D}}(W)). \tag{34}$$

Note that in Eq. (33) and Eq. (34), $W$ and $Z_{i,j}$s are independent, therefore $\mathbb{E}^{W}[\ell(W,Z_{i,j})] = R_{\mathcal{D}}(W)$. Also, we have $\text{Var}^{W}[\ell(W,Z_{i,j})] \le R_{\mathcal{D}}(W)(1 - R_{\mathcal{D}}(W))$ because Bernoulli random variable has the

largest variance among the $[0,1]$-bounded random variables with the same mean. Plugging Eq. (33) and Eq. (34) into Eq. (32) we obtain

$$\mathbb{E}[\sum_{i=1}^{k}\rho_i^{(k)}(U_j)\ell(W,Z_{i,j})]^2 \leq \frac{1}{k}\mathbb{E}[(k-1)A_1(k,R_{\mathcal{D}}(W))+A_2(k,R_{\mathcal{D}}(W))]. \tag{35}$$

We can upper bound the LHS of Eq. (35) by maximizing it over $R_{\mathcal{D}}(W)$ to obtain

$$\frac{\partial[(k-1)A_1(k,R)+A_2(k,R)]}{\partial R}=0 \Rightarrow R^{\star}=\frac{k^2+k-4}{2k^2-4k}.$$

We can plug the expression for $R^{\star}$ into Eq. (35) to get

$$\mathbb{E}[\sum_{i=1}^{k}\rho_i^{(k)}(U_j)\ell(W,Z_{i,j})]^2 \leq \frac{1}{k}\mathbb{E}[(k-1)A_1(k,R^{\star})+A_2(k,R^{\star})]$$

$$=\frac{k^3+7k^2-8k-16}{4(k^3-2k^2)}. \tag{36}$$

Then, plugging Eq. (36) into Eq. (25) yields

$$\inf_{\lambda\geq 0}\frac{\text{CMI}_{\mathcal{D}}^k(\mathcal{A})}{\lambda}+\frac{\exp(\frac{\lambda}{n})-\frac{\lambda}{n}-1}{\frac{\lambda}{n}}\frac{k^3+7k^2-8k-16}{4(k^3-2k^2)} \geq \text{EGE}_{\mathcal{D}}(\mathcal{A}). \tag{37}$$

Finally, the closed form solution of Eq. (37) is given by

$$\frac{\partial\left[\frac{\text{CMI}_{\mathcal{D}}^k(\mathcal{A})}{\lambda}+\frac{\exp(\frac{\lambda}{n})-\frac{\lambda}{n}-1}{\frac{\lambda}{n}}\frac{k^3+7k^2-8k-16}{4(k-2)k^2}\right]}{\partial\lambda}=0 \Rightarrow$$

$$\lambda^{\star}=n\mathfrak{W}\left((\frac{4(k^3-2k^2)\text{CMI}_{\mathcal{D}}^k(\mathcal{A})}{n(k^3+7k^2-8k-16)}-1)\exp(-1)\right)+n,$$

which is the desired result. $\qquad\square$

## D   Proofs of Section 3

*Proof of Theorem 3.1.* With $k=2$, recall from the introduction

$$\tilde{Z}^{(2)}=\begin{pmatrix}Z_{1,1} & \cdots & Z_{1,n} \\ Z_{2,1} & \cdots & Z_{2,n}\end{pmatrix}\sim\mathcal{D}^{\otimes 2n},$$

and $U=(U_1,\ldots,U_n)\in\{1,2\}^n$ where $U_i$s are IID, and the marginal distribution follows $U_i \sim$ Bern$\left(\frac{1}{2}\right)$ for $i\in[n]$. Furthermore, recall $S=\{Z_{U_1,1},\ldots,Z_{U_n,n}\}$. The expected generalization error can be written as

$$\mathbb{E}\left[R_{\mathcal{D}}(W)-\hat{R}_S(W)\right]=\mathbb{E}\left[\frac{1}{n}\sum_{i=1}^{n}(-1)^{U_i}(\ell(Z_{1,i},W)-\ell(Z_{2,i},W))\right] \tag{38}$$

$$=\mathbb{E}\left[\frac{1}{m}\sum_{i=1}^{m}(-1)^{U_{J_i}}(\ell(Z_{1,J_i},W)-\ell(Z_{2,J_i},W))\right], \tag{39}$$

where the last equality follows because $J$ is independent of $U_J,\tilde{Z}^{(2)}$, and $W$.

Define $\tilde{W}$, $\tilde{U}_J$, and $\tilde{J}$ such that $(W,U_J,J,\tilde{Z}^{(2)})\stackrel{\text{d}}{=}(\tilde{W},\tilde{U}_J,\tilde{J},\tilde{Z}^{(2)})$, and $\tilde{W}$, $\tilde{U}_J$, and $\tilde{J}$ are independent given $\tilde{Z}^{(2)}$. By the Donsker–Varadhan variational formula [5, Prop. 4.15] and the disintegration theorem [11, Thm. 6.4], for all measurable functions $g$ in $\mathcal{G}$, i.e., the class of all functions $g$ such that $\left(\mathbb{P}^{\tilde{Z}^{(2)}}[\tilde{W}]\otimes\mathbb{P}^{\tilde{Z}^{(2)}}[\tilde{U}_J]\otimes\mathbb{P}^{\tilde{Z}^{(2)}}[\tilde{J}]\right)(\exp g)<\infty$, with probability one we have

$$I^{\tilde{Z}^{(2)}}(W,J;U_J)=\text{KL}(\mathbb{P}^{\tilde{Z}^{(2)}}[W,J,U_J]\,\|\,\mathbb{P}^{\tilde{Z}^{(2)}}[\tilde{W}]\otimes\mathbb{P}^{\tilde{Z}^{(2)}}[\tilde{J}]\otimes\mathbb{P}^{\tilde{Z}^{(2)}}[\tilde{U}_J]) \tag{40}$$

$$=\sup_{g\in\mathcal{G}}\mathbb{P}^{\tilde{Z}^{(2)}}[W,J,U_J](g)-\log\left[\left(\mathbb{P}^{\tilde{Z}^{(2)}}[\tilde{W}]\otimes\mathbb{P}^{\tilde{Z}^{(2)}}[\tilde{J}]\otimes\mathbb{P}^{\tilde{Z}^{(2)}}[\tilde{U}_J]\right)(\exp g)\right]. \tag{41}$$

Define $f(w, j, u_j) \triangleq \frac{\lambda}{m} \sum_{i=1}^{m} (-1)^{u_{j_i}} \left( \ell\left(z_{1,j_i}, w\right) - \ell\left(z_{2,j_i}, w\right) \right)$ where $\lambda \geq 0$. By Eq. (41), we can write

$$I^{\tilde{Z}^{(2)}}(W,J;U_J) \geq \mathbb{P}^{\tilde{Z}^{(2)}}[W,U_J,J](f) - \log\left[ \left( \mathbb{P}^{\tilde{Z}^{(2)}}[\tilde{W}] \otimes \mathbb{P}^{\tilde{Z}^{(2)}}[\tilde{U}_J] \otimes \mathbb{P}^{\tilde{Z}^{(2)}}[\tilde{J}] \right) (\exp f) \right]. \qquad (42)$$

Considering the second term of the RHS of Eq. (42), Hoeffding's lemma implies that

$$\left( \mathbb{P}^{\tilde{Z}^{(2)}}[\tilde{W}] \otimes \mathbb{P}^{\tilde{Z}^{(2)}}[\tilde{U}_J] \otimes \mathbb{P}^{\tilde{Z}^{(2)}}[\tilde{J}] \right) (\exp f) \qquad (43)$$

$$= \mathbb{E}^{\tilde{Z}^{(2)}} \exp\left( \frac{\lambda}{m} \sum_{i=1}^{m} (-1)^{\tilde{U}_{J_i}} \left( \ell\left(Z_{1,\tilde{J}_i}, \tilde{W}\right) - \ell\left(Z_{2,\tilde{J}_i}, \tilde{W}\right) \right) \right) \qquad (44)$$

$$= \mathbb{E}^{\tilde{Z}^{(2)}} \mathbb{E}^{\tilde{Z}^{(2)}, \tilde{W}, \tilde{J}} \prod_{i=1}^{m} \exp\left( \frac{\lambda}{m} (-1)^{\tilde{U}_{J_i}} \left( \ell\left(Z_{1,\tilde{J}_i}, \tilde{W}\right) - \ell\left(Z_{2,\tilde{J}_i}, \tilde{W}\right) \right) \right) \qquad (45)$$

$$\leq \mathbb{E}^{\tilde{Z}^{(2)}} \mathbb{E}^{\tilde{Z}^{(2)}, \tilde{W}, \tilde{J}} \prod_{i=1}^{m} \exp\left( \frac{\lambda^2 \left( \ell\left(Z_{1,\tilde{J}_i}, \tilde{W}\right) - \ell\left(Z_{2,\tilde{J}_i}, \tilde{W}\right) \right)^2}{2m^2} \right) \qquad (46)$$

$$\leq \exp\left( \frac{\lambda^2}{2m} \right), \qquad (47)$$

where we use the fact that

$$\left( \mathbb{P}^{\tilde{Z}^{(2)}}[\tilde{W}] \otimes \mathbb{P}^{\tilde{Z}^{(2)}}[\tilde{U}_J] \otimes \mathbb{P}^{\tilde{Z}^{(2)}}[\tilde{J}] \right) (f) = 0.$$

Substituting the bound in Eq. (47) into Eq. (42), rearranging and taking expectations, we obtain

$$\mathbb{E} \frac{1}{m} \sum_{i=1}^{m} (-1)^{U_{J_i}} \left( \ell\left(Z_{1,J_i}, W\right) - \ell\left(Z_{2,J_i}, W\right) \right) \leq \mathbb{E} \inf_{\lambda \geq 0} \frac{I^{\tilde{Z}^{(2)}}(W,J;U_J) + \frac{\lambda^2}{2m}}{\lambda} \qquad (48)$$

$$= \mathbb{E} \sqrt{\frac{2}{m} I^{\tilde{Z}^{(2)}}(W,J;U_J)}. \qquad (49)$$

Moreover, we have a.s.

$$I^{\tilde{Z}^{(2)}}(J,W;U_J) - \underbrace{I^{\tilde{Z}^{(2)}}(J;U_J)}_{0} = I^{\tilde{Z}^{(2)}}(W;U_J|J). \qquad (50)$$

Here, $I^{\tilde{Z}^{(2)}}(J;U_J) = 0$ since $J$ is independent of $U_J$ given $\tilde{Z}^{(2)}$. Plugging Eq. (50) into Eq. (49), we obtain the desired result. $\qquad \square$

*Proof of Theorem 3.2.* Consider

$$I(W;U_{J^{(m_1)}}|\tilde{Z}^{(2)}, J^{(m_1)}) = H(U_{J^{(m_1)}}|J^{(m_1)}, \tilde{Z}^{(2)}) - H(U_{J^{(m_1)}}|J^{(m_1)}, \tilde{Z}^{(2)}, W) \qquad (51)$$

$$= \frac{1}{\binom{n}{m_1}} \sum_{K_1 \in [n]_{m_1}} H(U_{K_1}|\tilde{Z}^{(2)}) - H(U_{J^{(m_1)}}|J^{(m_1)}, \tilde{Z}^{(2)}, W) \qquad (52)$$

$$= \frac{1}{\binom{n}{m_1}} \sum_{K_1 \in [n]_{m_1}} H(U_{K_1}|\tilde{Z}^{(2)}) - \frac{1}{\binom{n}{m_1}} \sum_{K_1 \in [n]_{m_1}} H(U_{K_1}|\tilde{Z}^{(2)}, W). \qquad (53)$$

Eq. (52) follows because $\tilde{Z}^{(2)} \perp\!\!\!\perp J^{(m_1)}$, while Eq. (53) follows because the event $\{J^{(m_1)} = K_1\}$ is independent of $(W, U_{K_1}, \tilde{Z}^{(2)})$. Then

$$\frac{1}{m_1} I(W; U_{J^{(m_1)}} | \tilde{Z}^{(2)}, J^{(m_1)}) = \frac{1}{m_1 \binom{n}{m_1}} \sum_{K_1 \in [n]_{m_1}} [H(U_{K_1}) - H(U_{K_1} | W, \tilde{Z}^{(2)})] \tag{54}$$

$$= \frac{1}{n} H(U) - \frac{1}{m_1 \binom{n}{m_1}} \sum_{K_1 \in [n]_{m_1}} H(U_{K_1} | W, \tilde{Z}^{(2)}) \tag{55}$$

$$= \frac{1}{m_2 \binom{n}{m_2}} \sum_{K_2 \in [n]_{m_2}} H(U_{K_2}) - \frac{1}{m_1 \binom{n}{m_1}} \sum_{K_1 \in [n]_{m_1}} H(U_{K_1} | W, \tilde{Z}^{(2)}) \tag{56}$$

$$\leq \frac{1}{m_2 \binom{n}{m_2}} \sum_{K_2 \in [n]_{m_2}} [H(U_{K_2}) - H(U_{K_2} | W, \tilde{Z}^{(2)})] \tag{57}$$

$$= \frac{1}{m_2} I(W; U_{J^{(m_2)}} | \tilde{Z}^{(2)}, J^{(m_2)}). \tag{58}$$

Eq. (54) follows from Eq. (53) and the fact that $U \perp\!\!\!\perp \tilde{Z}^{(2)}$, while Eq. (55) follows from each element of $U$ being IID. Eq. (57) follows from Lemma F.1, which is a modified version of the Han's inequality [22]. Finally, the last step follows from using the same line of reasoning as in Eq. (51) to Eq. (53).

Having established Eq. (4), the claim follows from

$$\mathbb{E}\sqrt{\frac{2}{m} I^{\tilde{Z}^{(2)}}(W; U_J | J)} \leq \sqrt{\frac{2}{m} I(W; U_J | \tilde{Z}^{(2)}, J)} \tag{59}$$

$$\leq \sqrt{\frac{2}{n} I(W; U | \tilde{Z}^{(2)})}, \tag{60}$$

where Eq. (59) is Jensen's inequality, and Eq. (60) is the direct application of Eq. (4) with $m_1 = m$ and $m_2 = n$. This proves the desired result. $\qquad\square$

*Proof of Theorem 3.4.* By the Donsker–Varadhan variational formula [5, Prop. 4.15] and the disintegration theorem [11, Thm. 6.4], with probability one, for all measurable functions $g$ such that $\left(\mathbb{P}^{\tilde{Z}^{(2)}}[\tilde{W}] \otimes \mathbb{P}^{\tilde{Z}^{(2)}}[\tilde{U}_i]\right)(\exp g) < \infty$, we have

$$I^{\tilde{Z}^{(2)}}(U_i, W) = \text{KL}(\mathbb{P}^{\tilde{Z}^{(2)}}[U_i, W] \| \mathbb{P}^{\tilde{Z}^{(2)}}[\tilde{U}_i] \otimes \mathbb{P}^{\tilde{Z}^{(2)}}[\tilde{W}]) \tag{61}$$

$$\geq \mathbb{P}^{\tilde{Z}^{(2)}}[U_i, W][g(W, \tilde{Z}^{(2)}, U_i)] - \log \mathbb{P}^{\tilde{Z}^{(2)}}[\tilde{U}_i] \otimes \mathbb{P}^{\tilde{Z}^{(2)}}[\tilde{W}][\exp(g(\tilde{W}, \tilde{Z}^{(2)}, \tilde{U}_i))] \tag{62}$$

where $\left(W, U_i, \tilde{Z}^{(2)}\right) \stackrel{d}{=} \left(\tilde{W}, \tilde{U}_i, \tilde{Z}^{(2)}\right)$ and $\tilde{W} \perp\!\!\!\perp \tilde{U}_i \mid \tilde{Z}^{(2)}$. For $i \in [n]$, let

$$g_i\left(W, \tilde{Z}^{(2)}, U_i\right) \triangleq \lambda \left(-1\right)^{U_i} \left(\ell\left(Z_{1,i}, W\right) - \ell\left(Z_{2,i}, W\right)\right).$$

Hoeffding's lemma implies that

$$\mathbb{P}^{\tilde{Z}^{(2)}}[\tilde{U}_i] \otimes \mathbb{P}^{\tilde{Z}^{(2)}}[\tilde{W}][\exp(g_i(\tilde{W}, \tilde{Z}^{(2)}, \tilde{U}_i))] \leq \exp\left(\frac{\lambda^2}{2}\right), \tag{63}$$

where in the last line we have used $g_i \in [-\lambda, \lambda]$ a.s. From (62), we obtain

$$\mathbb{E}^{\tilde{Z}^{(2)}} (-1)^{U_i} \left(\ell\left(Z_{1,i}, W\right) - \ell\left(Z_{2,i}, W\right)\right) \leq \inf_{\lambda \geq 0} \frac{\text{KL}(\mathbb{P}^{\tilde{Z}^{(2)}}[U_i, W] \| \mathbb{P}^{\tilde{Z}^{(2)}}[U_i] \otimes \mathbb{P}^{\tilde{Z}^{(2)}}[W]) + \frac{\lambda^2}{2}}{\lambda} \tag{64}$$

$$= \sqrt{2 I^{\tilde{Z}^{(2)}}(W; U_i)}. \tag{65}$$

Then, averaging over $i$ and taking expectations,

$$\mathbb{E}\left[R_\mathcal{D}(W) - \hat{R}_S(W)\right] = \mathbb{E}\frac{1}{n} \sum_{i=1}^{n} \mathbb{E}^{\tilde{Z}^{(2)}} (-1)^{U_i} \left(\ell\left(Z_{1,i}, W\right) - \ell\left(Z_{2,i}, W\right)\right) \tag{66}$$

$$\leq \frac{1}{n} \sum_{i=1}^{n} \mathbb{E}\sqrt{2 I^{\tilde{Z}^{(2)}}(W; U_i)}. \tag{67}$$

$$\square$$

*Proof of Theorem 3.7.* For any two random measures $P(\tilde{Z}^{(2)}, U_{J^c}, J)$ and $Q(\tilde{Z}^{(2)}, U)$ on $\mathcal{W}$, the Donsker–Varadhan variational formula [5, Prop. 4.15] and the disintegration theorem [11, Thm. 6.4], give that with probability one

$$\text{KL}(Q(\tilde{Z}^{(2)}, U) \| P(\tilde{Z}^{(2)}, U_{J^c}, J)) = \sup_{g \in \mathcal{G}} \left( Q(\tilde{Z}^{(2)}, U)[g] - \log P(\tilde{Z}^{(2)}, U_{J^c}, J)[\exp g] \right) \quad (68)$$

where $\mathcal{G} = \{g : P(\tilde{Z}^{(2)}, U_{J^c}, J)(\exp g) < \infty\}$.

Let $g = \frac{\lambda}{m} \sum_{j \in J} (-1)^{U_j} (\ell(Z_{1,j}, W) - \ell(Z_{2,j}, W))$. First, note that

$$\mathbb{E}^{\tilde{Z}^{(2)}, U_{J^c}, J} \left[ \frac{\lambda}{m} \sum_{j \in J} (-1)^{U_j} (\ell(Z_{1,j}, W) - \ell(Z_{2,j}, W)) \right] = 0.$$

This is because $\{U_j\}_{j \in J}$ are independent of $\tilde{Z}^{(2)}, U_{J^c}$, and $J$. Moreover, $g$ is $[-\lambda, \lambda]$-bounded. Therefore, we can use the Hoeffding's lemma to obtain

$$\log P(\tilde{Z}^{(2)}, U_{J^c}, J)(\exp g) \leq \frac{\lambda^2}{2}.$$

Hence, from Eq. (68), we conclude that

$$Q(\tilde{Z}^{(2)}, U) \left[ \frac{1}{m} \sum_{j \in J} (-1)^{U_j} (\ell(Z_{1,j}, W) - \ell(Z_{2,j}, W)) \right]$$
$$\leq \inf_{\lambda > 0} \frac{\text{KL}(Q(\tilde{Z}^{(2)}, U, J) \| P(\tilde{Z}^{(2)}, U_{J^c}, J))}{\lambda} + \frac{\lambda}{2} = \sqrt{2\text{KL}(Q(\tilde{Z}^{(2)}, U) \| P(\tilde{Z}^{(2)}, U_{J^c}, J))}$$

almost surely. Finally, since $J \perp\!\!\!\perp \left( \tilde{Z}^{(2)}, U \right)$ we get

$$Q(\tilde{Z}^{(2)}, U) \left[ \frac{1}{m} \sum_{j \in J} (-1)^{U_j} (\ell(Z_{1,j}, W) - \ell(Z_{2,j}, W)) \right]$$
$$= Q(\tilde{Z}^{(2)}, U) \left[ \frac{1}{n} \sum_{i=1}^{n} (-1)^{U_j} (\ell(Z_{1,i}, W) - \ell(Z_{2,i}, W)) \right]$$
$$= \mathbb{E} \left[ R_{\mathcal{D}}(W) - \hat{R}_S(W) \right]$$

The desired result follows. □

# E Proofs of Section 4

*Proof of Theorem 4.2.* Considering the generalization bound in Theorem 3.7 and Lemma 3.6, we can write

$$\mathbb{E} \left[ R_{\mathcal{D}}(W) - \hat{R}_S(W) \right] \leq \mathbb{E} \sqrt{2\text{KL}(Q_T(S) \| P_T(\tilde{Z}^2, U_{J^c}, J))}$$
$$\leq \mathbb{E} \sqrt{\sum_{t=1}^{T} 2\mathbb{E}^{\tilde{Z}^{(2)}, U, J} \text{KL}(Q_{t|} \| P_{t|})}. \quad (69)$$

First, note that from Eq. (11) it follows that

$$Q_{t|} = \mathcal{N}(\mu_{Q_{t|}}, \frac{2\eta_t}{\beta_t} \mathbb{I}_d),$$

where the mean is given by

$$\mu_{Q_{t|}} =$$
$$W_{t-1} - \eta_{t-1} \frac{n-1}{n} \nabla \tilde{R}_{S_{J^c}}(W_{t-1}) - \frac{\eta_{t-1}}{n} \left( \mathbb{1}\{U_J = 1\} \nabla \tilde{\ell}(Z_{1,J}, W_{t-1}) + \mathbb{1}\{U_J = 2\} \nabla \tilde{\ell}(Z_{2,J}, W_{t-1}) \right).$$

Next, we propose the following construction of $P_{t|}$. Note that $P_{t|}$ is $\mathcal{F}_t$-measurable random probability measure where

$$\mathcal{F}_t = \sigma(S_{J^c}, Z_{1,J}, Z_{2,J}, J, W_{0:t-1}).$$

Hence we can exploit the information in the trajectory up to time $t$ to construct $P_{t|}$. In particular, we use the information in $\mathcal{F}_t$ to perform a binary hypothesis testing in which the two hypotheses are defined as

$$\mathcal{H}_1 : U_J = 1,$$
$$\mathcal{H}_2 : U_J = 2.$$

Equivalently, $\mathcal{H}_1$ and $\mathcal{H}_2$ can also be described as the hypotheses that $Z_{1,J}$ is a member of the training set and $Z_{2,J}$ is a member of the training set, respectively. Denote $\pi_t = (\pi_{t,1}, \pi_{t,2})$ as a probability vector whose $i-$th element shows the belief of the prior at time $t$ that the true hypothesis is $\mathcal{H}_i$ for $i \in \{1,2\}$. Then, we consider the prior as

$$P_{t|} = \mathcal{N}(\mu_{P_{t|}}, \frac{2\eta_{t-1}}{\beta_{t-1}}\mathbb{I}_d), \tag{70}$$

where

$$\mu_{P_{t|}} = W_{t-1} - \eta_{t-1}\frac{n-1}{n}\nabla\tilde{R}_{S_{J^c}}(W_{t-1}) - \frac{\eta_{t-1}}{n}\left(\pi_{t,1}\nabla\tilde{\ell}(Z_{1,J}, W_{t-1}) + \pi_{t,2}\nabla\tilde{\ell}(Z_{2,J}, W_{t-1})\right). \tag{71}$$

Here $\pi_1 = (\frac{1}{2}, \frac{1}{2})$. Then, we construct the the belief vector $\pi_t$ for $t \geq 2$ using the log-likelihood ratio as

$$\pi_t = \left(\theta\left(\log\frac{\mathbb{P}^{\mathcal{F}_t}[\mathcal{H}_1]}{\mathbb{P}^{\mathcal{F}_t}[\mathcal{H}_2]}\right), 1 - \theta\left(\log\frac{\mathbb{P}^{\mathcal{F}_t}[\mathcal{H}_1]}{\mathbb{P}^{\mathcal{F}_t}[\mathcal{H}_2]}\right)\right), \tag{72}$$

where $\theta : \mathbb{R} \to [0,1]$. Also, we might expect that the optimal $\theta$ satisfies $\theta(0) = \frac{1}{2}$, $\lim_{x\to\infty}\theta(x) = 1$, and $\lim_{x\to-\infty}\theta(x) = 0$.

Denote probability density function $\mathbb{P}^{\tilde{Z}^{(2)}, U_{J^c}, \mathcal{H}_k, W_0}[W_{1:t-1}]$ as $f_k(W_{1:t-1})$ for $k \in \{1,2\}$. Due to Markov structure of the update rule in Eq. (11), we have

$$f_k(W_{1:t-1}) =$$
$$\prod_{i=1}^{t-1}\left(\frac{\beta_{i-1}}{4\pi\eta_{i-1}}\right)^{\frac{d}{2}}\exp\left(-\frac{\beta_{i-1}\|W_i - W_{i-1} + \eta_{i-1}\frac{n-1}{n}\nabla\tilde{R}_{S_{J^c}}(W_{i-1}) + \frac{\eta_{i-1}}{n}\nabla\tilde{\ell}(Z_{k,J}, W_{i-1})\|^2}{4\eta_{i-1}}\right). \tag{73}$$

Here, Eq. (73) is obtained by the Markov property of the update rule in Eq. (11). Then, since the prior distribution on $\mathcal{H}_1$ and $\mathcal{H}_2$ is uniform, we have

$$\log\frac{\mathbb{P}^{\mathcal{F}_t}[\mathcal{H}_1]}{\mathbb{P}^{\mathcal{F}_t}[\mathcal{H}_2]} = \log\frac{f_1(W_{1:t-1})}{f_2(W_{1:t-1})} \tag{74}$$
$$= Y_{t,2} - Y_{t,1}, \tag{75}$$

where $Y_{t,1}$ and $Y_{t,2}$ are given by

$$Y_{t,1} \triangleq \sum_{i=1}^{t-1}\frac{\beta_{i-1}}{4\eta_{i-1}}\|W_i - W_{i-1} + \eta_{i-1}\frac{n-1}{n}\nabla\tilde{R}_{S_{J^c}}(W_{i-1}) + \frac{\eta_{i-1}}{n}\nabla\tilde{\ell}(Z_{1,J}, W_{i-1})\|^2,$$
$$Y_{t,2} \triangleq \sum_{i=1}^{t-1}\frac{\beta_{i-1}}{4\eta_{i-1}}\|W_i - W_{i-1} + \eta_{i-1}\frac{n-1}{n}\nabla\tilde{R}_{S_{J^c}}(W_{i-1}) + \frac{\eta_{i-1}}{n}\nabla\tilde{\ell}(Z_{2,J}, W_{i-1})\|^2. \tag{76}$$

Therefore, the belief vector is given by

$$\pi_t = \left(\theta\left((Y_{t,2} - Y_{t,1})\right), 1 - \theta\left((Y_{t,2} - Y_{t,1})\right)\right), \tag{77}$$

where $Y_{0,1} = Y_{0,2} = 0$ and for $t \geq 2$, $Y_{t,1}$ and $Y_{t,2}$ are given by Eq. (76). To conclude the proof, we obtain

$$\text{KL}(Q_T(S) \,\|\, P_T(\tilde{Z}^{(2)}, U_{J^c}, J)) \leq \sum_{t=1}^{T} \mathbb{E}^{\tilde{Z}^{(2)}, U, J} \text{KL}(Q_{t|} \,\|\, P_{t|}) \tag{78}$$

$$= \sum_{t=1}^{T} \mathbb{E}^{\tilde{Z}^{(2)}, U, J} \frac{\beta_{t-1} \eta_{t-1} \|(\mathbb{1}\{U_J = 1\} - \pi_{t,1}) \nabla \tilde{\ell}(Z_{1,J}, W_{t-1}) + (\mathbb{1}\{U_J = 2\} - \pi_{t,2}) \nabla \tilde{\ell}(Z_{2,J}, W_{t-1})\|^2}{4n^2} \tag{79}$$

$$= \sum_{t=1}^{T} \mathbb{E}^{\tilde{Z}^{(2)}, U, J} \frac{\beta_{t-1} \eta_{t-1} (\mathbb{1}\{U_J = 1\} - \pi_{t,1})^2 \|\nabla \tilde{\ell}(Z_{1,J}, W_{t-1}) - \nabla \tilde{\ell}(Z_{2,J}, W_{t-1})\|^2}{4n^2} \tag{80}$$

Finally, plugging Eq. (80) into Eq. (69), we get the desired result in Eq. (14). □

## F  Conditional Han's Inequality

**Lemma F.1.** *Let $(X_1, \ldots, X_n, Y)$ be $n+1$-dimensional random variable where $X_1, \ldots, X_N$ are discrete random variables. Then,*

$$\frac{1}{k \binom{n}{k}} \sum_{T \in [n]_k} H(X_T | Y)$$

*is decreasing in $k$.*

*Proof.* For notational convenience, let $\overline{H}_k(X_{[n]}|Y) = \frac{1}{\binom{n}{k}} \sum_{T \in [n]_k} H(X_T|Y)$. Note that if we manage to show that

$$\overline{H}_k(X_{[n]}|Y) - \overline{H}_{k-1}(X_{[n]}|Y) \leq \overline{H}_{k+1}(X_{[n]}|Y) - \overline{H}_k(X_{[n]}|Y), \tag{81}$$

then the result in Lemma F.1 follows. To show Eq. (81), we can write

$$H(X_1, \ldots, X_{k+1}|Y) + H(X_1, \ldots, X_{k-1}|Y)$$
$$= H(X_1, \ldots, X_k|Y) + H(X_{k+1}|X_1, \ldots, X_k, Y) + H(X_1, \ldots, X_{k-1}|Y) \tag{82}$$
$$\leq H(X_1, \ldots, X_k|Y) + H(X_{k+1}|X_1, \ldots, X_{k-1}, Y) + H(X_1, \ldots, X_{k-1}|Y) \tag{83}$$
$$= H(X_1, \ldots, X_k|Y) + H(X_1, \ldots, X_{k-1}, X_{k+1}|Y). \tag{84}$$

Here in Eq. (83), we drop $X_k$ from the condition in the second term. Therefore, we have

$$H(X_1, \ldots, X_{k+1}|Y) + H(X_1, \ldots, X_{k-1}|Y) \leq H(X_1, \ldots, X_k|Y) + H(X_1, \ldots, X_{k-1}, X_{k+1}|Y). \tag{85}$$

Then, by averaging Eq. (85) over all $n!$ permutation of $\{1, \ldots, n\}$, we get the desired result in Eq. (81). □

## G  Details of Experiments

In this section, we discuss the details behind the experiments as well as the details of minimizing the generalization bound in Theorem 4.2.

### G.1  Network architectures and learning curve

Tables 2 to 5 summarize the hyper-parameters we used for the experiments. Also, in Fig. 3 we plot the learning curves for the experiments reported in Section 4.2.1.

### G.2  Optimizing the bound over the choice of $\theta$ function

Our generalization bound in Theorem 4.2 consists of an infimum over the functions in $\Theta$. To study the impact of infimum, we consider the family of functions $\Theta$ given by

$$\Theta = \{\theta_a(x) | \exists a > 0 \text{ such that } \theta_a(x) = \frac{1}{2}(1 + \text{erf}(\frac{x}{a})) \text{ or } \theta_a(x) = \frac{1}{2}(1 + \tanh(\frac{x}{a}))\}.$$

(a) MNIST with MLP

(b) MNIST with CNN

(c) Fashion-MNIST

(d) CIFAR10

Figure 3: Learning curves. These plots show the training error, error on the test set, and the training loss. The loss functions is cross-entropy. Note y-axes for the error plots are log-scale.

| Dataset | MNIST |
|---|---|
| Architecture | MLP(784-500-500-10) |
| $\eta_t$ | $0.06 \times (0.95)^{\lceil \frac{t}{50} \rceil}$ |
| $\frac{2\eta_t}{\beta_t}$ | $10^{-8} + (3 \times 10^{-6} - 10^{-8}) \times \exp(-0.5 \lceil \frac{t}{50} \rceil)$ |
| Number of iterations | 900 |
| Final training error | $4.33 \pm 0.01\%$ |
| Generalization error | $0.88 \pm 0.01\%$ |
| Number of training examples | 20000 |
| Number of runs | 100 |

Table 2: Details of Experiments reported for MNIST with MLP

| Dataset | MNIST |
|---|---|
| Architecture | CL($5 \times 5(32)$)-MaxPool($2 \times 2$)-CL($5 \times 5(64)$) MaxPool($2 \times 2$)-FC(128)-FC(10) |
| $\eta_t$ | $0.05 \times (0.90)^{\lceil \frac{t}{40} \rceil}$ |
| $\frac{2\eta_t}{\beta_t}$ | $10^{-8} + (10^{-5} - 10^{-8}) \times \exp(-0.5 \lceil \frac{t}{40} \rceil)$ |
| Number of iterations | 700 |
| Final training error | $2.59 \pm 0.01\%$ |
| Generalization error | $0.55 \pm 0.01\%$ |
| Number of training examples | 20000 |
| Number of runs | 100 |

Table 3: Details of Experiments reported for MNIST with CNN

| Dataset | Fashion-FMNIST |
|---|---|
| Architecture | CL($5 \times 5(32)$)-MaxPool($2 \times 2$)-CL($5 \times 5(64)$) MaxPool($2 \times 2$)-FC(200)-FC(10) |
| $\eta_t$ | $0.07 \times (0.95)^{\lceil \frac{t}{50} \rceil}$ |
| $\frac{2\eta_t}{\beta_t}$ | $5 \times 10^{-8} + (7 \times 10^{-6} - 5 \times 10^{-8}) \times \exp(-0.3 \lceil \frac{t}{50} \rceil)$ |
| Number of iterations | 1300 |
| Final training error | $7.96 \pm 0.03\%$ |
| Generalization error | $3.71 \pm 0.03\%$ |
| Number of training examples | 20000 |
| Number of runs | 100 |

Table 4: Details of Experiments reported for Fashion-MNIST with CNN

| Dataset | CIFAR10 |
|---|---|
| Architecture | CL($3 \times 3(32)$)-MaxPool($2 \times 2$)-CL($3 \times 3(64)$) CL($3 \times 3(32)$)-MaxPool($2 \times 2$)-FC(128)-FC(10) |
| $\eta_t$ | $0.15 \times (0.98)^{\lceil \frac{t}{50} \rceil}$ |
| $\frac{2\eta_t}{\beta_t}$ | $10^{-9} + (3 \times 10^{-5} - 10^{-9}) \times \exp(-0.3 \lceil \frac{t}{50} \rceil)$ |
| Number of iterations | 2300 |
| Final training error | $9.39 \pm 0.46\%$ |
| Generalization error | $32.89 \pm 0.44\%$ |
| Number of training examples | 15000 |
| Number of runs | 100 |

Table 5: Details of Experiments reported for CIFAR10 with CNN

Then, we divide the samples of the optimization trajectory into two sets of equal size: training set and the test set. Then, we optimize over $a$ to find the $\theta_{a^\star}(x)$ that achieves the minimum expected generalization over the training set. The numbers reported in Table 1 are based on the evaluation of $\theta_{a^\star}(x)$ over the test set. Thus, the number reported in Table 1 are unbiased estimate of the generalization bound in Theorem 4.2.