[Reviews · NeurIPS 2020]

Review 1

Summary and Contributions: Roughly, the major contribution of the manuscript is twofold: (1) Authors provide a refined information-based generalization bound. (2) Authors apply the bound to analyze the learning algorithm based on Langevin dynamics. Authors also provide some results regarding relationship between MI and CMI (Section 2) and some prior-based scenarios, which are relatively minor compared to the other results.

Strengths: I quite enjoyed reading this manuscript, mainly due to the comprehensiveness of the analysis. Having provided a refined generalization bound, the authors step forward to analyze the prior-based scenarios and Langevin dynamics-based learning procedures. The numerical results in section 4.2.1 were also helpful for understanding the current state of theory-practice gap on generalization of neural network models. Additionally, - the manuscript is nicely structured, - the relevant literatures are adequately (and honestly) cited, - while the main results look deceptively simple, the authors have taken sufficient effort to concretely establish the claims.

Weaknesses: - I think the paper could have been written more clearly (in my humble opinion). For instances: (1) IOMI is could have been more "properly," making it easier for the readers to locate. (2) Remark 3.5 states that the last inequality follows from the independence of indices U_i, which took several lines for me to verify formally. (3) Lemma 3.6. can be more clarified by stating exact which result of [15] the authors refer to and how it was adapted. Similar issue with Theorem 1.1 and Lemma 4.1 (there could be more). - The results in Section 2 are either expected, or not sufficiently discussed. Theorem 2.1. is probably well-expected (in my opinion) to the readers who are familiar of the work of Steinke and Zakynthinou. Regarding Theorem 2.2., I must admit that I failed to understand the utility of the bound, especially given the strict "finiteness" assumption. If CMI bounds are better, why is it useful to know that it can recover MI bounds? Is there any specific case where MI bound is easily analyzable and CMI bound is not? - I hate to say this, but the main ideas underlying the proof are not strikingly new.

Correctness: Yes; I mostly focused on verifying the results in Section 3. Could have missed some subtle issues.

Clarity: It is well-structured, but there are some clarity issues here and there; see above for more details.

Relation to Prior Work: Yes.

Reproducibility: Yes

Additional Feedback: Thank you for the response. ==================== Nitpicks here. - I suggest writing "Definition 1.2." instead of "Def. 1.2.," given that the authors write "Theorem" instead of "Thm." - What is \mathcal{W}^{0,...,T}, in line 177? Was it defined properly? - Typo, line 182, then -> the - Is subscript of expectation identical to superscript? (Lemma 4.1.)


Review 2

Summary and Contributions: This paper deals with upper bounds on the expected generalization gap (EGE) of classification algorithms, specifically looking at data-dependent bounds (i.e. that can be estimated from the training data itself). It builds on the recent work of Russo and Zou who proved that this quantity can be upper bounded using the mutual information between the output of the algorithm and the training sample (IOMI), and Steinke and Zakynthinou who showed that this quantity can be upper bounded by the conditional mutual information between the output of the algorithm and a selection function conditioned on a super-sample (CMI^k). This work first shows that CMI^k is always smaller than IOMI and converges to IOMI as k goes to infinity, and then provides a refined bound where the CMI is generalized to use a random subset of the selection function. This new bound is then applied to the Langevin dynamics algorithm (leveraging the fact that one can use a prior that depends on optimization trajectory) to derive an improved generalization bound for this algorithm which is shown empirically to have a smaller numerical value than previous bounds.

Strengths: Soundness: - theoretical grounding: the proofs seem to be correct to the extent I could verify. - empirical evaluation: the experiments seem reasonable and appear to support the claim of the improved tightness of the proposed bound Significance and novelty: Relevance: Other comments:

Weaknesses: Relevance: One important question is whether the proposed bounds lead to new insights. The empirical demonstration of obtaining a tighter bound for the Langevin dynamics algorithm is certainly a good first step to show that the new data-dependent quantity that appears in the bounds captures partially how favorable the distribution is for the algorithm, but it is hard to say that one can derive from that an improved understanding of the algorithm itself. Also it is not clear whether this could lead to designing an even better algorithm. So I believe that the question of "what can be gained from the new bound" is not really addressed, which is why my score is not higher.

Correctness: No issue here

Clarity: No issue here, the paper is very readable.

Relation to Prior Work: The connections and differences with prior work are clearly articulated.

Reproducibility: Yes

Additional Feedback:


Review 3

Summary and Contributions: I'd like to begin with the disclaimer that I'm only superficially familiar with information-theoretic generalization bounds. However I'm quite familiar with other learning-theoretic tools and the area of generalization bounds in deep learning. ----------------- Short summary --------------- This paper builds on a line of work studying information-theoretic generalization bounds. The paper first begins by relating the recent Conditional Mutual Information-based bounds by S&Z'20 to the Mutual Information-based bounds by R&Z'16/X&R'17. Next, the paper establishes general, tighter CMI-based bounds inspired by other recent ideas from Negrea et al., '19 & Bu et al., '19. Finally, these bounds are instantiated for the Langevin Dynamics algorithm where these bounds are then empirically demonstrated to be tighter. More details: ----------------- a) The paper first shows that the CMI term in S&Z'20 is always <= the mutual information term in R&Z'16/X&R'17. They also show an equality when the super-sample size used in CMI tends to infinity. b) Next, inspired by Bu et al., '19., the paper presents a more general version of the S&Z'20. In S&Z'20, one computes I( W; U | Z) where Z is the super-sample of n+n datapoints and U consists of n 0/1 values corresponding which half of Z each of the datapoints in the training set S come from. This paper's bound is based on I(W; U_{J} | Z, J) where J is a random subset of {1, ... n} of cardinality m (and when m=n, we get the original bound). This general version is shown to be tighter than S&Z'20; the tightest bound achieved when m=1. c) In order to make the above bound practically computable, the paper employs an idea from Negrea et al., '19 to bound the CMI in terms of the KL divergence between a prior and posterior. -- This form of the bound also gives some idea as to why this paper's bound is tighter than what you'd get from a KL-version of the original CMI bound: the prior in this bound can be data-dependent in that it can be "informed by" the n-m datapoints in S not indexed by J (i.e., S_{J^c}). d) Finally, the paper applies this KL divergence bound for noisy, iterative algorithms. Like is standard in this line of work, the KL divergence term is split into a chain of KL divergence terms corresponding to each step of the iteration. A crucial contribution here is to cleverly design the the prior for each of these steps for the specific LD algorithm, within the above framework. --- In more detail: under the setting where m=1, the paper presents a prior for any time t as a convex combination of the two different weights that one would reach starting from W_{t-1} under the two possible choices for the J'th datapoint. The exact ratio of this combination is "informed by" looking W_1, W_2 ... W_{t-1} and gauging which of the two choices in the supersample is really in the training set. e) The above bound is numerically computed for a variety of settings (MNIST+MLP, MNIST+CNN, F-MNIST+CNN, CIFAR+CNN), and is shown to be non-vacuous compared to the bounds of Negrea et al., '19 and Li et al., '20. Also, interestingly, these bounds saturate with the number of timesteps unlike the other existing bounds (which is also what was intuitively expected from the fact that the priors in later timesteps can be more informed.)

Strengths: 1. The paper has built on a multitude of recent ideas to advance our understanding of information-theoretic generalization bounds. 2. Although these ideas are inspired by recent papers, it is just as valuable to identify which existing ideas can be combined, how to combine them and actually show when the combinations works and in what aspects it is better. 3. Overall, this exploration has resulted in multiple solid, valuable and novel contributions ranging from foundational ones (reg. the CMI bounds, and the new generalization bound) to more empirical/applied ones (instantiating the bound for LD, and showing that it's better than existing bounds). 4. I appreciate the fact that the paper presents "a complete story" in that it presents a general tool and also applies it. 5. I think it's also exciting to see that empirically the resulting bound saturates with iteration count, unlike existing bounds which increase as we train the network further and further. This is an important aspect that one'd want from a generalization bound; further, it seems like the idea of fixing a super-sample and focusing on a subset of it, is critical to achieving this.

Weaknesses: I don't have any negative feedback on this paper. I think this is a nice paper definitely worth publishing. 6. I do have some suggestions regarding the clarity (noted under the question on clarity), but I must add some of those might simply be because of my lack of expertise in information-theoretic generalization bounds.

Correctness: 7. The paper seems quite rigorous. However, unfortunately, I don't have the expertise to carefully check the arguments in a short period of time.

Clarity: The abstract and introduction is crystal clear about the contributions. There are some parts within the text that I found under-explained, that I enumerate here: 8. While combining the KL divergence idea with the CMI bound of this paper in Eq 9, it's not clear how the U_{J_c} found a place in the mutual information term I(W; U_J | U_{J_c}, J). A direct application of the bound would only result in I(W; U_J | J) right? I think I understand why this is ok (because the CMI increases if we condition it on U_{J_c} since it's independent of U_{J}). Is this correct? A yes or (no+short explanation) would suffice! Either ways, it'd be great to be clear about it on the paper. -- Side note: The mapping between the CMI-KL-divergence lemma (Lemma 3.6) and "what the prior can know" is a key concept. This could be explained more elaborately and more informally. (see also point 17) 9. The description of the prior could be clearer e.g., "Prior takes an LD step" didn't initially make sense to me because the prior is a joint-distribution which doesn't change with time. I later realized this is talking about constructing the conditional distribution P(W_t| W_{t-1}, ... W_0) for the t'th time step based on the t-1'th step. 10. It'd be very important to add some intuition regarding the terms in Equations in 12, 13, 14 and 15. In general, it's better if the reader is given some verbal intuition regarding cumbersome mathematical expressions. Otherwise, it is quite time-consuming to unpack these terms. 11. Importantly, it'd be helpful to emphasize to the reader as to how Eq 14 captures the foreshadowed "exploitation of information from the optimization trajectory to identify U". 12. I'd also consider re-arranging the order in which the terms Y_{t,u}, \Theta etc., are introduced.

Relation to Prior Work: 13. Yes. I love that the paper is honest about its novelty, and gives due credit to the ideas it builds on (really appreciate all the citations in the abstract).

Reproducibility: No

Additional Feedback: =============== Updates after response: Thank you for responding to all the questions within the short page limit. The responses clarify my questions. As requested I have increased my confidence score by a point. =============== I only have some minor clarification questions: 14. If I understand everything correctly, even the prior in Negrea et al., '19 becomes more and more informed with the timestep (as it can depend on W_1 ... W_{t-1}). And despite this, their bound grows with timesteps, while the bound here doesn't. What explains this discrepancy? Does this boil down to the fact that in this paper the prior (a) is given access to a super-sample Z (b) and it only needs to figure out a subset of S by searching through Z, while on the other hand, in Negrea et al., the prior has to figure out a subset of S without access to any such Z? 14. Even though Thm 3.2 implies that we get the tightest CMI bound when the subset size m=1, does this still hold for the KL-divergence based upper bound? Is it right to say that this may not necessarily be the case? 15. If it's indeed the case that m=1 may not be optimal for the KL-divergence upper bound, one would naturally be curious about numerically evaluating the given bound for larger values of m. But the catch seems to be that there's no obvious way to extend the given idea for "identifying U" to larger m. When m > 1, there'll be exponentially many different possible values for U_{J} and so it's not clear how one would identify them (perhaps, we could approximate it in some way?). 16. Is there a reason why the plots do not show the actual (expected) generalization error? ========= 17. My only other major, friendly suggestion would be to be aware of the fact that the paper could benefit a wider (NeurIPS) audience if it could be made more accessible. The rigor in the mathematical language used by the paper is commendable and rare, but this could severely come in the way of an expert in another line of work who is a layperson here, and wants to understand and borrow from the ideas in this paper. For comparison, I'd say that I found S&Z'20 to be very accessible. On the other hand, in this paper, I think the intense and dense mathematical notation occlude all the great ideas. I admit it might be challenging to achieve these somewhat conflicting goals of being simple and accessible vs., being rigorous in exposition. Two concrete suggestions: a) Using the appendix more resourcefully (either provide longer layperson-friendly explanations there or do the opposite i.e., relegate rigorous statements to the appendix, while keeping the main paper simple) b) Reserve the rigorous language to only the theorem statements, while be more laid-back in the verbal explanations. Also, verbally express the theorem statement in informal language. Since there's no limit on the appendix length, the proofs in the appendix could also be padded with informal statements wherever possible, while preserving the formal statements as they are. This of course might just be due to personal stylistic differences or my lack of expertise in this line of work. So this doesn't affect my score in any way. ===== Minor typos: Line 17 by a -> be a Line 116: a refined bounds based In line 179 the notation of using Q in the subscript is not explained before. Line 240 csntant


Review 4

Summary and Contributions: This paper is along the recent very popular line of work on information theoretic generalization error bound. This paper first characterizes the relationship between two types of mutual information based generalization bounnd, and then proposed their own bound based on the idea of individual sample in [6] and the data dependent idea in [15]. Application to noisy and iterative algorithms are investigated. Numerical results are provided to validate their claims. I have read the authors' response. My concerns are properly addressed, and therefore I raised my score to 7.

Strengths: This paper exploits ideas from [6] and [15] and developed a tighter generalization error bound.

Weaknesses: 1. Theorem 1.1 ignores the subgaussian parameter in [18] and [24]. Same for theorem 1.2. 2. In section 2, the CMI and IOMI are comapred. Is it meaningful to compare only these two quantities? Since they may appear in the genearliation bound with different coefficient before them. 3. One generic question is that how do we exactly use these generalization bounds in practice to guide algorithm design? 4. When applied to neural networks with a large number of parameters, how to compute the bound (or how to estimate the bound)? In this high-dimensional case, most mutual information estimator will fail. If a bound is not computable, how can we use it in practice?

Correctness: it looks correct to this reviewer.

Clarity: yes

Relation to Prior Work: yes

Reproducibility: Yes

Additional Feedback:

[Author Response · NeurIPS 2020]

We thank all reviewers for their time and valuable comments. Below we address each review in turn.

**Reviewer 1 :** We incorporated your feedback on the presentation and added more explanation within the proofs.
Significance of the results in Section 2: The purpose of Section 2 is to unify the two main information-theoretic
approaches for studying generalization. [21] gave instances where the gap between CMI / IOMI was large, but did
not provide a general result. Thm. 2.1 shows $\mathrm{CMI}_{\mathcal{D}}^{k}(\mathcal{A})$ is a tighter measure of dependence than $\mathrm{IOMI}_{\mathcal{D}}(\mathcal{A})$ for *any*
*learning algorithm* and *any data distribution*. This is an important, novel inequality, even if it admits a straightforward
proof. We also address the role of the size of the super-sample in CMI. In [21], CMI is defined using a super-sample of
size $2n$ ($k = 2$) only. Thm. 2.2 addresses this by showing $\mathrm{CMI}_{\mathcal{D}}^{k}(\mathcal{A})$ and $\mathrm{IOMI}_{\mathcal{D}}(\mathcal{A})$ agree in the limit as $k \to \infty$.
We conjecture the finiteness assumption is an artifact of our analysis, though we don't have a more general proof.
Novelty of the proof techniques: Assuming this refers to Section 2 (as our new bounds for Langevin dynamics represent
such a clear and material advance), we believe our results in Section 2 are essential for understanding the relationship
between CMI and IOMI, irrespective of the novelty of the proof techniques. That said, we think there are several fun
and subtle arguments buried in the proofs of Section 2.

**Reviewer 2 :** Insights from the LD bound: A prevailing method for analyzing the generalization error for iterative
algorithms is via the chain rule for KL, using priors for the joint distribution of weight vectors that are Markov, i.e.,
given the $t$th weight, the $(t + 1)$th weight is conditionally independent from the trajectory so far. Existing results using
this approach accumulate a "penalty" for each step. In [14, 15, 16] the penalty terms are (resp.) the squared norm of the
gradients, the trace of the gradient covariance, and the squared Lipschitz constant. The penalty term in our paper is the
squared norm of "two-sample incoherence", defined as the squared norm of the difference between the gradient of a
randomly selected training point and the held-out point. However, the use of chain rule along with existing "Markovian"
priors introduces a source of looseness, i.e., the accumulating penalty may diverge to $+\infty$ yielding vacuous bounds (as
seen in Fig. 1). The distinguishing feature of our data-dependent CMI analysis is that the penalty terms get "filtered" by
the online hypothesis test via our *non*-Markovian prior, i.e., our prediction for $t + 1$ depends on whole trajectory. When
the true index can be inferred from the prev. weights, then the penalty essentially stops accumulating. For instance it can
be seen in the middle column of Fig. 1 that the penalty term of our paper is close to or larger than [15]'s. Nevertheless,
the online hypothesis test discounts our penalties, yielding non-vacuous bounds where earlier bounds fails to.
Designing better algorithms: We're aware of work using PAC-Bayes bounds as training objectives. A similar approach
based on $\mathrm{CMI}_{\mathcal{D}}^{k}(\mathcal{A})$ is an avenue for future work, though use of data-dependent priors presents new challenges.

**Reviewer 3 :** We thank the reviewer for their exceptionally in-depth review. As the reviewer engaged with the material
at a high level, we believe it would be reasonable for Rev. 3 to increase their confidence score.
8. Regarding the first step in Eq. 9, the answer to your question is "yes". We will clarify this in the paper.
9.–12., 17. We included more intuition and explanations, and adopted your suggestions regarding the presentation.
14.(1) The prior in [15] doesn't exploit the optimization trajectory; it's a Markov process. Our prior is not Markov; it
"learns" the identity of the held out data point from the history. This is an important difference between the bounds.
14.(2) Recall the variational representation $I(X; Y) = \inf_P \mathbb{E}[\mathrm{KL}(Q(X)\|P)]$, where the infimum achieved by $P =$
$\mathbb{E}[Q(X)]$. If we always use the "optimal" prior in Eq. 3, we have "KL-based" bounds are tightest when $m = 1$.
However, since the priors for different values of $m$ are probability kernels with different input spaces ($J$ is a subset
of size $m$), we cannot compare the bound for a fixed prior as $m$ varies as it is impossible to have the same prior for
different $m$. Also note that to compute optimal prior we would need to know the data distribution.
15. For $m > 1$, the problem could either be reduced to a single selection between $2^m$ different alternatives, or to $m$
separate binary tests and a union bound. We consider this to be potential future work.
16. Fig. 2 in App. G plots the test and training error. We will move the plots to the body in the final version.

**Reviewer 4 :** 1. As stated in Sec. 1, our focus is on $[0, 1]$-valued loss functions. In Thm. 1.1, we translate the
sub-Gaussian results from [18] and [24] to bounded loss. Thm. 1.3, however, is taken from [21], which also does
not provide generalization bounds for Sub-Gaussian losses. While extensions to sub-Gaussian losses are often trivial,
extensions of the bounds in Section 3 are not immediate due to our particular use of boundedness. 2. Thm B.1 shows
that asymptotically, as the super-sample size increases, the leading coefficients can be taken to be the same. Also, the
$\mathrm{CMI}_{\mathcal{D}}^{k}(\mathcal{A})$ and $\mathrm{IOMI}_{\mathcal{D}}(\mathcal{A})$ play the same role in their respective bounds, capturing the dependence of the bound on
the complexity of the learned hypothesis, and it's dependence on the training data relative to the sample size.
3. Please see response to Reviewer 2's second question. 4. Perhaps this is a misunderstanding, but, in Sec. 4, we study
the generalization error of the Langevin dynamics algorithm as an example of noisy, iterative algorithms using the
bounds proposed in Section 3. We apply our bounds to large, overparametrized neural networks. We estimate an upper
bound on the mutual information via the KL between two Gaussians, to sidestep computational intractability of MI.
We use a data-dependent technique to estimate the mutual information in order to obtain tighter and easily simulated
bounds. In Sec. 4 we evaluate our proposed bound numerically and find that it is non-vacuous across various neural
network architectures and datasets where # networks parameters $\gg$ # training points. Finally, for the case of Lipschitz
loss function, we show in Remark 4.6 that our bound does not depend on the number of parameters.

[Meta-Review · NeurIPS 2020]

This paper makes progress on a recent sequence of papers on information-theoretic quantities in generalization bounds for statistical learning, in particular focusing on the mutual information and conditional mutual information, and providing a new (sometimes tighter) variant. The reviewers all agree that the progress made in this paper is valuable and worthy of publication, and note that the application to studying Langevin dynamics is a particularly nice component of the paper. However, they also note that the paper could have been written more accessibly.